# TMEM25 inhibits monomeric EGFR-mediated STAT3 activation in basal state to suppress triple-negative breast cancer progression

Jing Bi[1,11], Zhihui Wu[1,11], Xin Zhang[1,2,11], Taoling Zeng[1,11], Wanjun Dai[1], Ningyuan Qiu[1], Mingfeng Xu[1], Yikai Qiao[1], Lang Ke[1], Jiayi Zhao[1], Xinyu Cao[1], Qi Lin[1], Xiao Lei Chen [3,4], Liping Xie[4], Zhong Ouyang[5], Jujiang Guo[6], Liangkai Zheng[6], Chao Ma[7], Shiying Guo[8], Kangmei Chen[9], Wei Mo [1], Guo Fu [3,4,6] ✉, Tong-Jin Zhao [10] ✉ & Hong-Rui Wang [1,6] ✉

Triple-negative breast cancer (TNBC) is a subtype of breast cancer with poor outcome and lacks of approved targeted therapy. Overexpression of epidermal growth factor receptor (EGFR) is found in more than 50% TNBC and is suggested as a driving force in progression of TNBC; however, targeting EGFR using antibodies to prevent its dimerization and activation shows no significant benefits for TNBC patients. Here we report that EGFR monomer may activate signal transducer activator of transcription-3 (STAT3) in the absence of transmembrane protein TMEM25, whose expression is frequently decreased in human TNBC. Deficiency of TMEM25 allows EGFR monomer to phosphorylate STAT3 independent of ligand binding, and thus enhances basal STAT3 activation to promote TNBC progression in female mice. Moreover, supplying TMEM25 by adeno-associated virus strongly suppresses STAT3 activation and TNBC progression. Hence, our study reveals a role of monomeric-EGFR/STAT3 signaling pathway in TNBC progression and points out a potential targeted therapy for TNBC.

Epidermal growth factor receptor (EGFR, also known as ErbB1, HER1) is the prototype of the ErbB/HER family of receptor tyrosine kinase (RTKs) that controls various developmental processes via diverse signaling pathways including phosphatidylinositol 3-kinase (PI3K)/Akt, Ras/Raf/MEK/ERK, and STAT3 pathways[1]. EGFR remains monomeric and its kinase domain is auto-inhibited in the absence of ligand. In response to ligand such as EGF binding, EGFR and its family members,

ErbB2/HER2, ErbB3/HER3, and ErbB4/HER4, can form homo- or heterodimers, resulting in an allosteric conformational change to activate the kinase domain[2–4]. Meanwhile, the kinase domain can also be activated by an increased local concentration-induced dimerization of EGFR[2]. It has been long established that hyperactivation of EGFR signaling, due to point mutations, intragenic deletions, or overexpression, is closely associated with development of a variety of

[1]State Key Laboratory of Cellular Stress Biology, School of Life Sciences, Faculty of Medicine and Life Sciences, Xiamen University, 361102 Fujian, China. [2]State Key Laboratory of Proteomics, National Center for Protein Sciences (Beijing), Beijing Institute of Lifeomics, 100850 Beijing, China. [3]Cancer Research Center of Xiamen University, 361102 Xiamen, Fujian, China. [4]School of Medicine, Xiamen University, 361102 Fujian, China. [5]Department of Breast Surgery, The First Affiliated Hospital of Xiamen University, 361003 Xiamen, Fujian, China. [6]Department of Obstetrics and Gynecology, Women and Children's Hospital, School of Medicine, Xiamen University, 361003 Xiamen, Fujian, China. [7]Medical School of Chinese PLA, 100853 Beijing, China. [8]GemPharmatech Co., Ltd., 210000 Nanjing, Jiangsu, China. [9]Department of Clinical Laboratory, Sun Yat-sen Memorial Hospital, Sun Yat-sen University, 510120 Guangzhou, China. [10]Shanghai Key Laboratory of Metabolic Remodeling and Health, Institute of Metabolism and Integrative Biology, Zhongshan Hospital, Fudan University, 200438 Shanghai, China. [11]These authors contributed equally: Jing Bi, Zhihui Wu, Xin Zhang, Taoling Zeng. ✉e-mail: guofu@xmu.edu.cn; zhaotj@fudan.edu.cn; wanghr@xmu.edu.cn

human cancers[5,6]. Among them, EGFR is overexpressed in more than 50% of triple-negative breast cancer (TNBC), which is characterized by lack of estrogen receptor, progesterone receptor, and HER2 expression, associated with poor clinical outcome and has limited treatment approaches[7–9]. It has been widely believed that dimerization is an essential step for activation of EGFR. Intriguingly, clinical phase trials using EGFR monoclonal antibody cetuximab to block EGFR dimerization did not show statistically significant efficacy in patients with TNBC[10–12].

Signal transducer and activator of transcription (STAT) family proteins are transcription factors that regulate a wide range of gene transcription in response to various cytokine and growth factor signaling pathways[13]. Among the STAT family members, STAT3 and STAT5 are well known for their roles in promoting cancer progression, especially STAT3, which is currently considered as a promising target for cancer therapy[14–16]. Activation of STATs is commonly via phosphorylation of a single conserved tyrosine residue in the C-terminal domain, which induces dimerization and subsequent nuclear translocation of STATs[13]. STATs can be activated through various signaling pathways. Canonically, receptors without intrinsic kinase activity such as cytokine receptors for interleukin-6 (IL-6) recruits the Janus kinases (JAKs) and STATs to cytoplasmic domain of the receptor, where JAKs

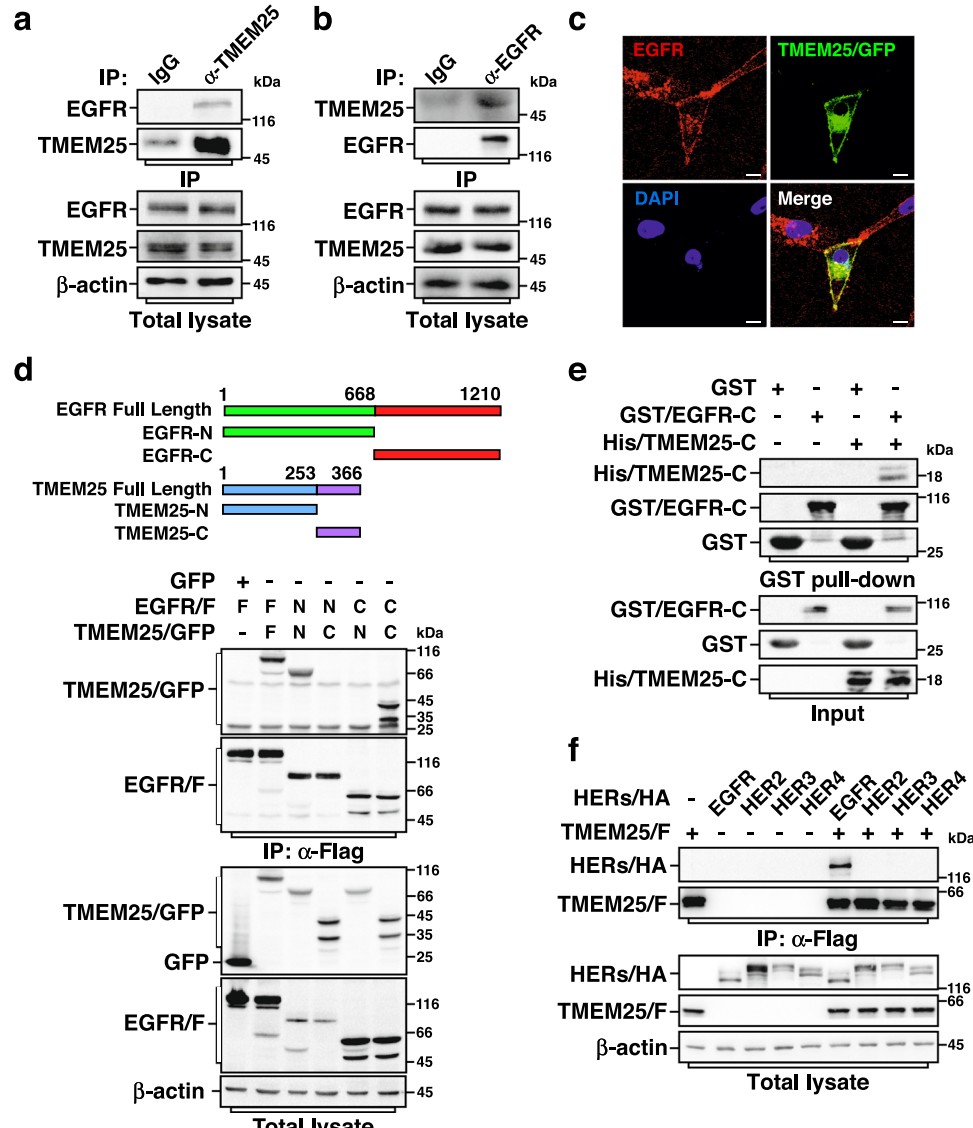

**Fig. 1 | TMEM25 interacts with EGFR. a, b** TMEM25 interacts with EGFR endogenously. Cell lysates from MDA-MB-231 cells were subjected to anti-TMEM25 IP followed by immunoblotting to detect associated EGFR (**a**) or to anti-EGFR IP followed by immunoblotting to detect associated TMEM25 (**b**). **c** TMEM25 colocalizes with EGFR at plasma membrane. MDA-MB-231 cells transfected with C-terminal green fluorescent protein (GFP)-tagged TMEM25 (TMEM25/GFP) were subjected to immunofluorescence assay to detect localization of TMEM25/GFP (green) and endogenous EGFR (red). The nuclei were stained with DAPI (blue). The scale bars indicate 10 μm. **d** TMEM25 interacts with EGFR via both extracellular domain and cytosolic domain. HEK293T cells transfected with indicated combinations of full length (F), N-terminal extracellular/transmembrane domains (N), or cytosolic domain (C) of TMEM25/GFP and C-terminal Flag-tagged EGFR (EGFR/F) were subjected to anti-Flag IP followed by immunoblotting assay to detect associated TMEM25. **e** Direct interaction between cytosolic domains of TMEM25 and EGFR. Bacterially expressed and purified His-tagged TMEM25 cytosolic domain (His/TMEM25-C) and GST-tagged EGFR cytosolic domain (GST/EGFR-C) were subjected to GST pull-down assay. **f** TMEM25 interacts with EGFR but not HER2, HER3, or HER4. HEK293T cells transfected with indicated combinations of C-terminal hemagglutinin (HA)-tagged EGFR, HER2, HER3, or HER4 (HERs/HA) and C-terminal Flag-tagged TMEM25 (TMEM25/F) were subjected to anti-Flag IP followed by immunoblotting assay to detect associated HERs. The blotting experiments are representative of at least three biologically independent replicates (**a, b, d–f**). Source data are provided as a Source Data file.

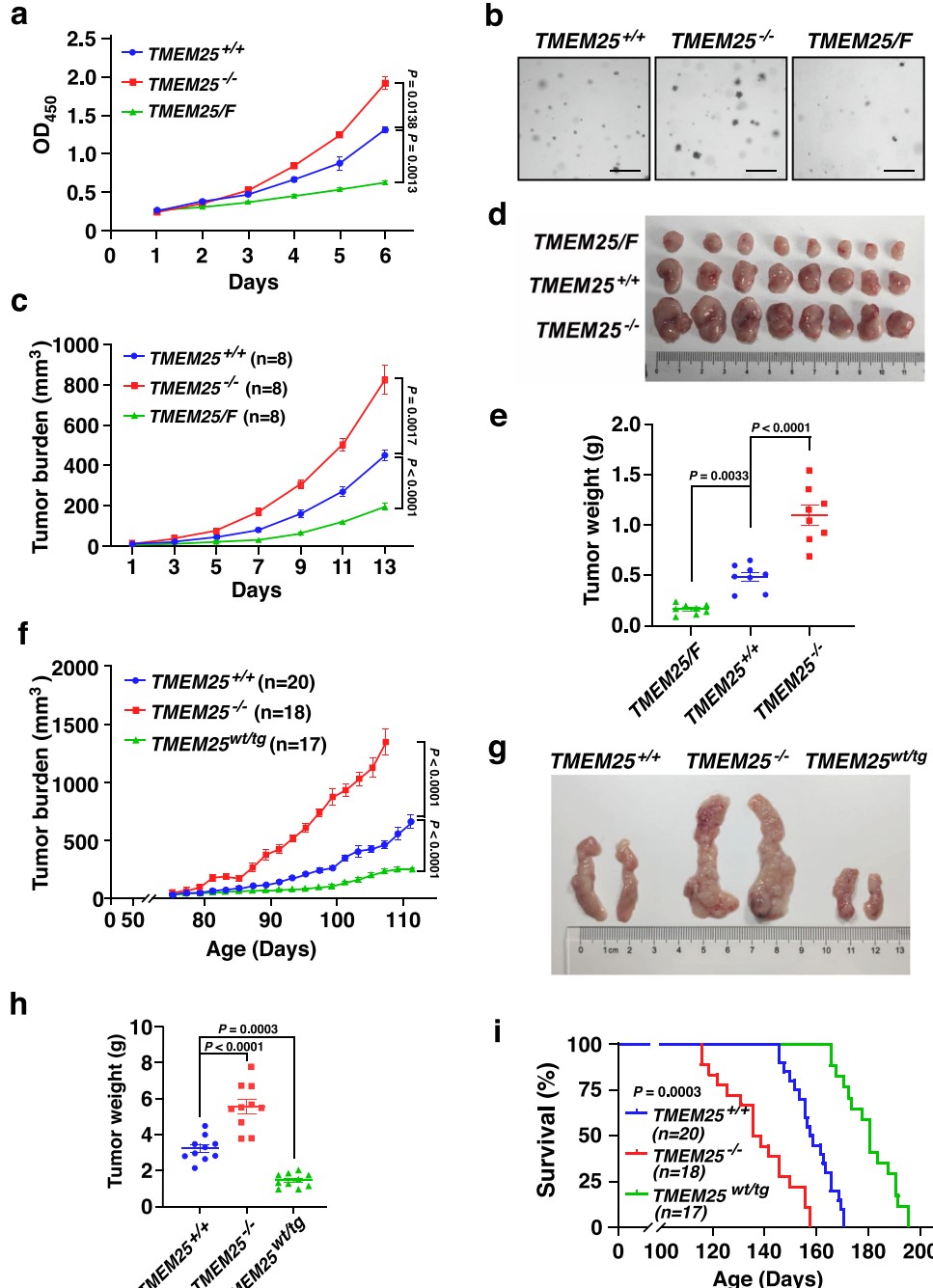

**Fig. 2 | TMEM25 inhibits TNBC progression. a** Effect of TMEM25 on growth of MDA-MB-231 cells. Cell viability of wild-type (*TMEM25⁺/⁺*), *TMEM25* knockout (*TMEM25⁻/⁻*), and Flag-tagged TMEM25 overexpressing (*TMEM25/F*) MDA-MB-231 cells were determined using CCK-8 assay. Data were presented as mean ± SEM of three individual experiments. *P* values were determined by two-way ANOVA followed by Tukey test. **b** Effect of TMEM25 on colony formation of MDA-MB-231 cells. *TMEM25⁺/⁺*, *TMEM25⁻/⁻*, and *TMEM25/F* MDA-MB-231 cells were subjected to soft agar assay. Scale bar indicates 100 μm. At least three biological replicates were performed for each condition. **c–e** TMEM25 suppresses xenograft tumor growth of MDA-MB-231 cells in nude mice. *TMEM25⁺/⁺*, *TMEM25⁻/⁻*, or *TMEM25/F* MDA-MB-231 cells were orthotopically injected into the mammary fat pad of nude mice to observe tumor formation. Tumor volumes were measured and plotted as mean ± SEM (*n* = 8 animals per group). *P* values were determined by two-way ANOVA followed by Tukey test (**c**). The tumors were obtained 13 days after tumor formation by sacrificing the mice (**d**), weighted and plotted as mean ± SEM (*n* = 8 animals per

group). *P* values were determined by one-way ANOVA followed by Tukey test (**e**). **f–h** TMEM25 suppresses spontaneous TNBC tumor growth in *MMTV-PyMT* mice. Tumor volumes in *TMEM25⁺/⁺*, *TMEM25⁻/⁻*, or *TMEM25* transgenic (*TMEM25ʷᵗ/ᵗᵍ*) *MMTV-PyMT* mice were measured at same ages and plotted as mean ± SEM of a pool of indicated number of mice per group. *P* values were determined by two-way ANOVA followed by Tukey test (**f**). Representative tumors were obtained from one *TMEM25⁺/⁺*, *TMEM25⁻/⁻*, or *TMEM25ʷᵗ/ᵗᵍ MMTV-PyMT* mouse at age of 100 days (**g**). Tumors obtained from 100 days aged *TMEM25⁺/⁺*, *TMEM25⁻/⁻* and *TMEM25ʷᵗ/ᵗᵍ MMTV-PyMT* mice were weighted and plotted as mean ± SEM (*n* = 10 animals per group). *P* values were determined by one-way ANOVA followed by Tukey test (**h**). **i** Effect of TMEM25 on survival of *MMTV-PyMT* mice. Overall survival rates of *TMEM25⁺/⁺*, *TMEM25⁻/⁻*, or *TMEM25ʷᵗ/ᵗᵍ MMTV-PyMT* mice in (**f**) were determined. *P* values were determined by log-rank (Mantel-Cox) test, 95% confidence interval of ratio. Source data are provided as a Source Data file.

are activated to phosphorylate STATs[14]. In addition, RTKs such as EGFR and platelet-derived growth factor receptor (PDGFR) can also mediate STATs activation by either directly phosphorylating STATs or indirectly through recruiting non-receptor tyrosine kinases such as SRC[13,17]. STAT3 is often constitutively activated in all classes of breast tumors, but most frequently in TNBC[18,19].

Although overexpression of EGFR and hyperactivation of STAT3 are associated with poor prognosis of TNBC patients[8,19], little is known whether and how these two events are associated. To date, numerous efforts have been made to target STAT3 pathway for cancer treatment. However, although dozens of inhibitors targeting STAT3 or its upstream pathways have been tested in clinical trials, only a few are approved by the United States Food and Drug Administration (FDA) for treatment of cancers including pancreatic cancer and gastric/gastroesophageal junction cancer, but not TNBC yet. One of the major obstacles is the toxic side effects caused by ubiquitous expression of STAT3[20]. Therefore, finding a way to specifically target activation of STAT3 in cancer cells could be an appealing strategy.

Transmembrane (TMEM) protein family is a group poorly described proteins that contain one or more transmembrane domains spanning biological membranes[21]. TMEM25 is a single transmembrane protein, and was previously reported as a favorable prognostic and predictive marker for breast cancer patients; however, the molecular mechanism is unclear[22]. In addition, the expression of TMEM25 was found decreased in colorectal cancer[23]. Recent study showed that TMEM25 modulates neuronal excitability by affecting NMDA receptor NR2B subunit degradation[24].

In this study, we identify TMEM25 as an EGFR binding protein to prevent monomeric EGFR-mediated phosphorylation of STAT3 in the basal state, therefore functioning as a tumor suppressor to inhibit TNBC progression. Our study also demonstrates a potential of TMEM25 for targeted therapy of TNBC, and suggests that clinical mutations of TMEM25 identified in other types of cancer may be able to trigger monomeric-EGFR/STAT3 signaling to promote tumor progression as well.

## Results

### TMEM25 interacts with EGFR and its expression is decreased in TNBC

To study the underlying mechanism of EGFR in promoting TNBC development, we performed a yeast-two-hybrid screen using EGFR as a bait and identified TMEM25 as a binding protein of EGFR. To verify this interaction in cells, we performed co-immunoprecipitation (Co-IP) assays using either exogenously expressed or endogenous EGFR and TMEM25, confirming that EGFR could indeed interact with TMEM25 (Fig. 1a, b and Supplementary Fig. 1a, b). Meanwhile, using the C-terminal green fluorescent protein (GFP)-tagged TMEM25 (TMEM25/GFP), we determined that endogenous EGFR and TMEM25/GFP were mainly colocalized at plasma membrane by immunofluorescence assay (Fig. 1c). We next carried out domain mapping assay and found that EGFR and TMEM25 interact with each other through both their N-terminal extracellular domains and C-terminal cytosolic domains (Fig. 1d). In addition, we performed GST pull-down assay and detected an interaction between bacterially expressed EGFR cytosolic domains and TMEM25 cytosolic domain in vitro (Fig. 1e), suggesting that the cytosolic domains of EGFR and TMEM25 might be able to directly interact with each other. Furthermore, we compared the interactions between TMEM25 and HER family proteins and found that TMEM25 specifically interacts with EGFR but not HER2/3/4 (Fig. 1f and Supplementary Fig. 1c).

Previous studies reported that expression of TMEM25 is decreased in breast cancer and colorectal cancer[22,23]. We analyzed the datasets from the cancer genome atlas (TCGA) using the UALCAN platform (http://ualcan.path.uab.edu/analysis.html) and found that mRNA levels of TMEM25 was indeed decreased in colorectal cancers compared with normal colorectal tissues; however, we did not see a significant difference in survival rate in colorectal cancer patients with different expression levels of TMEM25 (Supplementary Fig. 2a, b). Interestingly, in breast cancers, TMEM25 mRNA levels were decreased most dramatically in TNBCs compared with normal mammary tissues (Supplementary Fig. 2c). In agreement of the report that TMEM25 is a favorable marker for survival of breast cancer patient[22], analysis using UCSC Xena platform (https://xena.ucsc.edu) showed a statistically significant difference of survival rates in breast cancer patients with relatively high and low expression of TMEM25 (Supplementary Fig. 2d). Therefore, we next focused our study on investigating the role of TMEM25 in TNBC progression.

### TMEM25 inhibits growth of TNBC cells in vitro and in vivo

To investigate the role of TMEM25 in regulating TNBC progression, we first examined the impact of TMEM25 expression on growth rates of TNBC cells using human TNBC MDA-MB-231 cells and murine TNBC 4T1 cells. Remarkably, in both cell culture and soft agar colony formation assays, knockout of TMEM25 significantly promoted the growth of these cells, whereas overexpression of TMEM25 dramatically inhibited the cell growth (Fig. 2a, b and Supplementary Fig. 3a, b). The protein levels of TMEM25 in these cells were confirmed by immunoblotting assay (Supplementary Fig. 3c). Accordingly, knockout of TMEM25 significantly increased the xenograft tumor growth of MDA-MB-231 cells in nude mice whereas overexpression of TMEM25 inhibited the tumor growth (Fig. 2c–e). Similar results were also obtained from transplanted tumor formation experiment using 4T1 cells in BALB/c mice (Supplementary Fig. 3d–f). The expression levels of TMEM25 in the tumors were confirmed by immunoblotting assay (Supplementary Fig. 3g). Meanwhile, the expression of Ki67, a molecular marker of proliferating cells, in the tumors were examined by immunohistochemistry assay (Supplementary Fig. 3h), indicating that knockout or overexpression of TMEM25 indeed drastically enhanced or inhibited cell proliferation in the tumors, respectively.

### TMEM25 inhibits growth of spontaneous TNBC in mice

To further evaluate the role of TMEM25 in progression of TNBC, we generated TMEM25 knockout (TMEM25[−/−]) mice by deleting exons 2 to 5 of TMEM25 (Supplementary Fig. 4a) and transgenic (TMEM25[wt/tg]) mice using the piggyBac transposon system[25] (Supplementary Fig. 4b). The relative TMEM25 mRNA levels of the TMEM25[−/−] and TMEM25[wt/tg] mice were confirmed by real-time quantitative PCR (Supplementary Fig. 4c). We then crossed TMEM25[−/−] or TMEM25[wt/tg] mice with the mouse mammary tumor virus LTR-driven polyoma middle T antigen (MMTV-PyMT) transgenic mice, which is characterized as an androgen receptor-positive TNBC model[26], to examine the function of TMEM25 in regulating spontaneous TNBC development. Strikingly, knockout of TMEM25 drastically promoted the tumor growth and lung metastasis, whereas overexpression of TMEM25 markedly suppressed the tumor growth and lung metastasis (Fig. 2f–h and Supplementary Fig. 4d–f). It is noteworthy that the normalized lung metastasis to primary tumor burden was also increased by knockout of TMEM25 and decreased by overexpression of TMEM25 (Supplementary Fig. 4f). Accordingly, the overall survival time was significantly shortened in TMEM25[−/−] mice and prolonged in TMEM25[wt/tg] mice (Fig. 2i). The expression levels of TMEM25 and Ki67 in the spontaneous tumors were also examined by immunoblotting and immunohistochemistry assays, respectively (Supplementary Fig. 4g, h). Similar as that in transplanted tumors (Supplementary Fig. 3h), knockout of TMEM25 promoted cell proliferation in the spontaneous tumors, whereas overexpression of TMEM25 suppressed the cell proliferation (Supplementary Fig. 4h). Furthermore, we confirmed that TMEM25 interacted with EGFR in tumors formed in MMTV-PyMT mice as well (Supplementary Fig. 4i). Hence, these results clearly demonstrated that TMEM25 plays a pivotal role in suppressing TNBC progression.

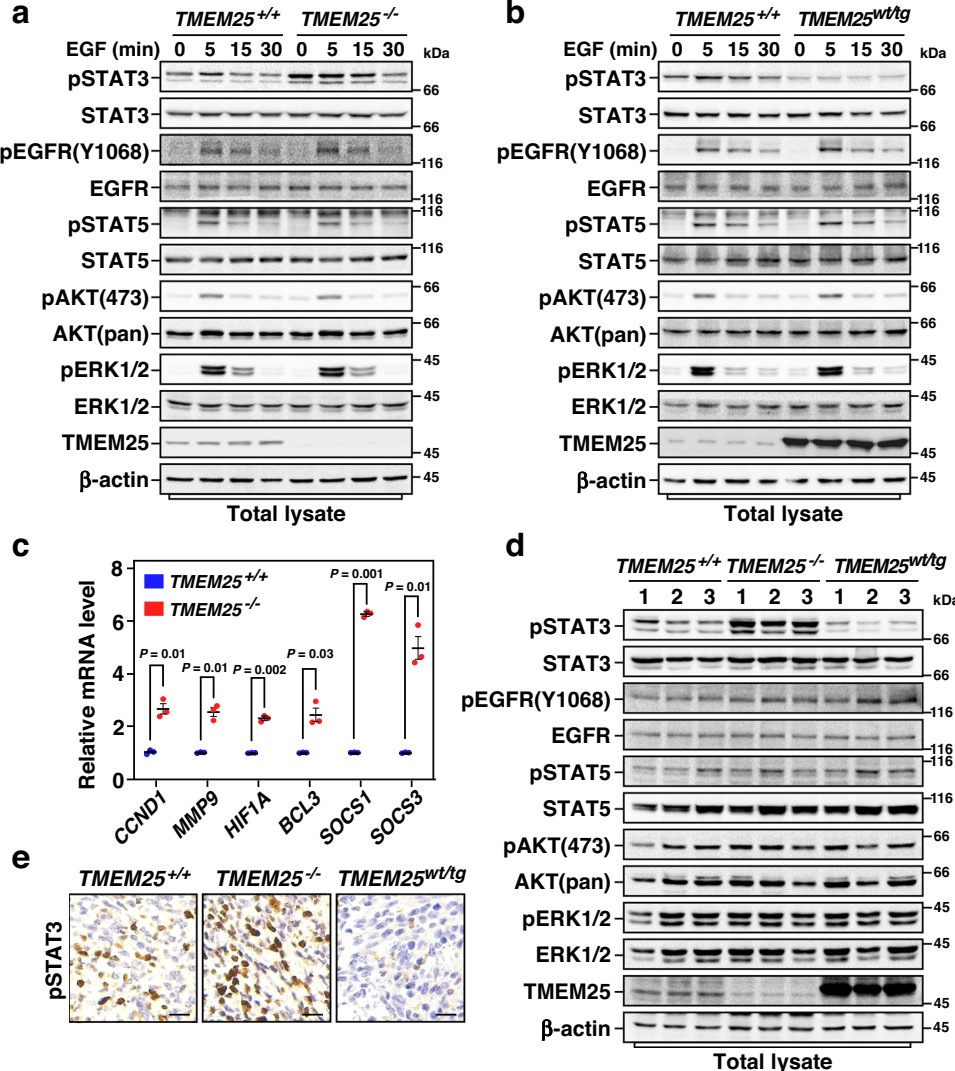

**Fig. 3 | TMEM25 negatively regulates STAT3 signaling. a** Knockout of *TMEM25* induces phosphorylation of STAT3 in MEF cells. Primary MEF cells derived from *TMEM25*$^{+/+}$ or *TMEM25*$^{-/-}$ mice were serum-starved overnight and then treated with 100 ng/ml EGF for the indicated time before subjected to immunoblotting assay. **b** Overexpression of TMEM25 inhibits EGF treatment-induced phosphorylation of STAT3 in MEF cells. Primary MEF cells derived from *TMEM25*$^{+/+}$ or *TMEM25*$^{wt/tg}$ mice were treated and analyzed as in (**a**). **c** Knockout of *TMEM25* increases mRNA levels of representative STAT3 target genes. The mRNA levels of indicated genes in *TMEM25*$^{+/+}$ or *TMEM25*$^{-/-}$ MDA-MB-231 cells were measured by real-time quantitative PCR using *GAPDH* as an internal control. Data are presented as mean ± SEM of three independent experiments. *P* values were determined by two-tailed unpaired Student's *t* test. **d, e** TMEM25 inhibits STAT3 phosphorylation in the spontaneous breast tumors in *MMTV-PyMT* mice. Representative tumors from *TMEM25*$^{+/+}$, *TMEM25*$^{-/-}$, or *TMEM25*$^{wt/tg}$ *MMTV-PyMT* mice were subjected to immunoblotting assay (**d**) or IHC assay (**e**) as indicated. Scale bar indicates 100 μm. The blotting experiments are representative of at least three biologically independent replicates (**a, b, d**). Source data are provided as a Source Data file.

## TMEM25 inhibits STAT3 signaling

Next, we examined the effects of TMEM25 on regulating EGFR signaling pathways using primary mouse embryonic fibroblast (MEF) cells derived from *TMEM25*$^{-/-}$ or *TMEM25*$^{wt/tg}$ mice. Interestingly, neither knockout nor overexpression of TMEM25 affected EGF treatment-induced phosphorylation of EGFR (Y1068), ERK (T202/Y204), or AKT (S473) in these cells (Fig. 3a, b). However, phosphorylation of STAT3 (Y705) but not STAT5 (Y694), a close family member of STAT3, in *TMEM25*$^{-/-}$ cells was strikingly enhanced even without EGF treatment (Fig. 3a). Consistent overexpression of TMEM25 substantially attenuated EGF treatment-induced phosphorylation of STAT3 (Fig. 3b), suggesting a specific role of TMEM25 in regulating STAT3 signaling, especially preventing activation of STAT3 in the basal state. Knockout of *TMEM25* resulted in phosphorylation of STAT3 in the basal state when neither EGFR were auto-phosphorylated nor AKT and ERK signaling pathways were activated (Fig. 3a), suggesting that TMEM25

depletion-induced phosphorylation of STAT3 is not via Ras/MAPK or PI3K/AKT pathway.

We further confirmed the effects of knockout or overexpression of TMEM25 on regulating the phosphorylation levels of STAT3 in human TNBC MDA-MB-231, BT549, and HCC1937 cells, and murine TNBC 4T1 cells (Supplementary Fig. 5a, b). It is noteworthy that ERK signaling pathway in MDA-MB-231 cells and both AKT and ERK signaling pathways in 4T1 cells are constitutively activated; however, knockout or overexpression of TMEM25 showed similar effects on regulating STAT3 phosphorylation in these cells as that in primary MEF cells (Supplementary Fig. 5a, b), further indicating that AKT and ERK are not involved in TMEM25-regulated STAT3 signaling pathway. We performed quantitative reverse transcription polymerase chain reaction (qRT-PCR) analysis of *CCND1, MMP9, HIF1A, BCL3, SOCS1*, and *SOCS3*, which are known STAT3 target genes in breast cancer development[27–29], and found that knockout of *TMEM25* increased transcription of these genes

(Fig. 3c), indicating that TMEM25 indeed functions as an inhibitor of STAT3 signaling. We confirmed that phosphorylation levels of STAT3 were markedly increased in the spontaneous tumors obtained from *TMEM25*$^{-/-}$-*MMTV-PyMT* mice and decreased in tumors from *TMEM25*$^{wt/tg}$-*MMTV-PyMT* mice, whereas no significant differences in phosphorylation levels of EGFR, AKT, ERK, or STAT5 were observed (Fig. 3d). The phosphorylation levels of STAT3 in spontaneous and transplant tumors were also examined by immunohistochemistry assays (Fig. 3e and Supplementary Fig. 5c), further confirming that the phosphorylation levels of STAT3 were increased in *TMEM25*$^{-/-}$ tumors and decreased in TMEM25 overexpressing tumors compared with that in tumors formed by wild-type cells. In agreement with previous report that STAT3 signaling enhances tumor metastasis by promoting epithelial-to-mesenchymal transition[30], knockout of TMEM25 also increased the lung metastasis rate of TNBC in *MMTV-PyMT* mice (Supplementary Fig. 4f).

### TMEM25 suppresses EGFR-mediated phosphorylation of STAT3

We were greatly interested in the activation of STAT3 in the basal state without treatment of ligand (Fig. 3a and Supplementary Fig. 5a). To investigate by which kinase STAT3 was phosphorylated in *TMEM25*$^{-/-}$ cells in the basal state, we first examined the requirement of JAK1/2, SRC, and EGFR in this phosphorylation of STAT3 by knocking down these kinases under the condition of serum starvation. Interestingly, knockdown of EGFR strongly suppressed the *TMEM25* knockout-induced enhancement of STAT3 phosphorylation, whereas knockdown of JAK1/2 or SRC showed less effect than that for knockdown of EGFR (Fig. 4a, b), suggesting that *TMEM25* knockout-induced phosphorylation of STAT3 is mainly dependent on EGFR activity. Similar results were also obtained by comparing the effects of treatment with EGFR kinase inhibitors (gefitinib or erlotinib), JAK1/2 inhibitors (ruxolitinib or baricitinib), or SRC inhibitors (bosutinib or saracatinib) (Supplementary Fig. 6a, b).

We next evaluated the necessity of TMEM25 in controlling STAT3 phosphorylation under different serum conditions. Remarkably, knockout of *TMEM25* yielded a similar amount of increase in the phosphorylation levels of STAT3 in all different concentrations of fetal bovine serum (FBS), and treatment of EGFR inhibitors totally blocked *TMEM25* knockout-induced increase of STAT3 phosphorylation (Fig. 4c and Supplementary Fig. 6c), indicating that *TMEM25* knockout has a general effect on increasing EGFR/STAT3 signaling in all serum conditions. Interestingly, although the increase amount of STAT3 phosphorylation induced by EGFR is similar under different serum conditions, it is noteworthy that its percentage to the total STAT3 phosphorylation was dramatically higher in low serum conditions, suggesting that this EGFR-mediated STAT3 phosphorylation may has a more potent role for cells in low serum conditions. Indeed, knockout of TMEM25 not only promoted cell proliferation in moderate or high concentrations of serum (5–10% FBS), but also was critical for cell survival in low serum conditions (≤2.5% FBS). Treatment of EGFR inhibitors or knockout of *EGFR* completely blocked the *TMEM25* knockout-promoted cell growth or survival (Fig. 4d and Supplementary Fig. 6d). Moreover, treatment of STAT3 inhibitor or knockdown of STAT3 totally blocked the *TMEM25* knockout-promoted cell survival (Supplementary Fig. 6e), and introduction of constitutively active mutant STAT3-Y705E fully rescued the cell survival that was inhibited by treatment of EGFR inhibitors or knockout of *EGFR* in *TMEM25*$^{-/-}$ cells under serum starvation (Supplementary Fig. 6f, g). Furthermore, knockdown of STAT3 drastically suppressed xenograft tumor growth of both wild-type and *TMEM25*$^{-/-}$ MDA-MB-231 cells in nude mice to a similar level (Supplementary Fig. 7a–e). Hence, these results suggested that loss of TMEM25 promotes cell growth or survival via activating EGFR/STAT3 signaling pathway, which may play a pivotal role in TNBC development, especially for the cells inside a tumor where with limited supply of growth factors.

In line with the specific interaction between TMEM25 and EGFR (Fig. 1f and Supplementary Fig. 1c), only introduction of EGFR but not HER2/3/4 into EGFR knockdown cells could rescue *TMEM25* knockout-induced phosphorylation of STAT3 (Fig. 4e), indicating that TMEM25 specifically inhibits EGFR/STAT3 signaling. Accordingly, overexpression of TMEM25 substantially blocked the interaction between STAT3 and EGFR in a dose-dependent manner in cells (Fig. 4f), and knockout of *TMEM25* markedly enhanced the interaction between endogenous EGFR and STAT3 (Fig. 4g). We verified that there was a direct interaction between cytosolic domain of EGFR (EGFR-C) and STAT3 using bacterially expressed proteins, and cytosolic domain of TMEM25 (TMEM25-C) could block the interaction between STAT3 and EGFR-C in a dose-dependent manner in vitro (Fig. 4h). Furthermore, we reconstituted the phosphorylation reaction of STAT3 in vitro using bacterially produced STAT3 and Flag-tagged EGFR affinity purified from HEK293T cells (Fig. 4i), which is in good agreement with previous report that STAT3 can be directly phosphorylated by EGFR in vitro[31]. In line with the binding assays (Fig. 4f–h), presence of TMEM25-C attenuated EGFR-mediated phosphorylation of STAT3 in a dose-dependent manner (Fig. 4i), suggesting that TMEM25 prevents EGFR-mediated phosphorylation of STAT3 by competitively binding to the cytosolic domain of EGFR.

### EGFR monomer phosphorylates STAT3 in the absence of TMEM25

It is well known that EGFR needs to bind ligand to form an asymmetric dimer to initiate its kinase activity for canonical EGFR signaling pathways[32]. Because EGFR could phosphorylate STAT3 under the condition of serum starvation in the absence of TMEM25 (Fig. 3a and Supplementary Fig. 5a), it hinted that the activity of EGFR toward STAT3 might be activated in a ligand-independent manner. To elucidate this, we used a reported dimerization-defective mutant EGFR-V948R to investigate how EGFR phosphorylates STAT3 in the absence of ligand in TMEM25 deficient cells. The V948 is located at the active dimer interface, which is critical for the asymmetric EGFR dimer formation[2, 3, 33]. Consistent with previous report[2], we confirmed that exogenously overexpressed wild-type EGFR could form dimer, whereas EGFR-V948R could not (Fig. 5a). To eliminate potential influence of endogenous EGFR, we further knocked out *EGFR* in *TMEM25*$^{-/-}$ cells and reintroduced Flag-tagged wild-type EGFR, EGFR-V948R, or kinase dead mutant EGFR-D837N[34] as a negative control into *EGFR*$^{-/-}$ *TMEM25*$^{-/-}$ cells. As shown in Fig. 5b, reintroduction of wild-type EGFR triggered AKT phosphorylation, whereas EGFR-D837N or EGFR-V948R did not. ERK is constitutively activated in MDA-MB-231 cells because these cells harbor K-RAS G13D and B-RAF G464V oncogenic mutations. Accordingly, phosphorylated Y1068 and Y1086, which are critical for EGFR-mediated activation of MAPK/ERK pathway[35, 36], PI3K/AKT pathway[37], and were suggested as docking sites for recruitment of STAT3 to EGFR[38], could only be detected in wild-type EGFR but not in EGFR-D837N or EGFR-V948R. However, in contrast to EGFR-D837N that could not rescue the STAT3 phosphorylation, EGFR-V948R showed a remarkable effect on rescuing the STAT3 phosphorylation as wild-type EGFR did, although its efficacy was to some extent lower than that of wild-type EGFR (Fig. 5b and Supplementary Fig. 8a), suggesting that, in the absence of TMEM25, the phosphorylation of STAT3 by EGFR only requires the kinase activity of EGFR but not necessarily dimer formation or phosphorylation of Y1068 and Y1086. Indeed, EGFR-V948R could interact with STAT3 as well (Fig. 5c), and both wild-type and V948R EGFR could phosphorylate STAT3 in vitro (Fig. 5d).

We next specifically examined the effects of phosphorylation of Y1068 and Y1086 in EGFR on EGFR-mediated STAT3 phosphorylation using EGFR-Y1068F, EGFR-Y1086F, and EGFR-Y1068,1086F (2YF) mutants. In line with the previous report[38], the 2YF mutation remarkably impeded binding of STAT3 to EGFR in the presence of TMEM25; however, it showed much less effect on attenuating the interaction between STAT3 and EGFR in the absence of TMEM25 (Supplementary Fig. 8b). Accordingly, although phosphorylation of Y1068 and Y1086

may affect the binding of EGFR to STAT3 and the efficacy of EGFR to phosphorylate STAT3 to some extent, EGFR-Y1068F, EGFR-Y1086F, and EGFR-2YF still could significantly enhance phosphorylation of STAT3 in *TMEM25*[−/−] cells (Fig. 5e and Supplementary Fig. 8c) and

in vitro (Fig. 5f). Moreover, we examined the global tyrosine phosphorylation of EGFR-V948R by immunoprecipitation followed by immunoblotting with anti-phospho-Tyr antibody pY20. Wild-type EGFR showed a basal tyrosine phosphorylation and EGF treatment

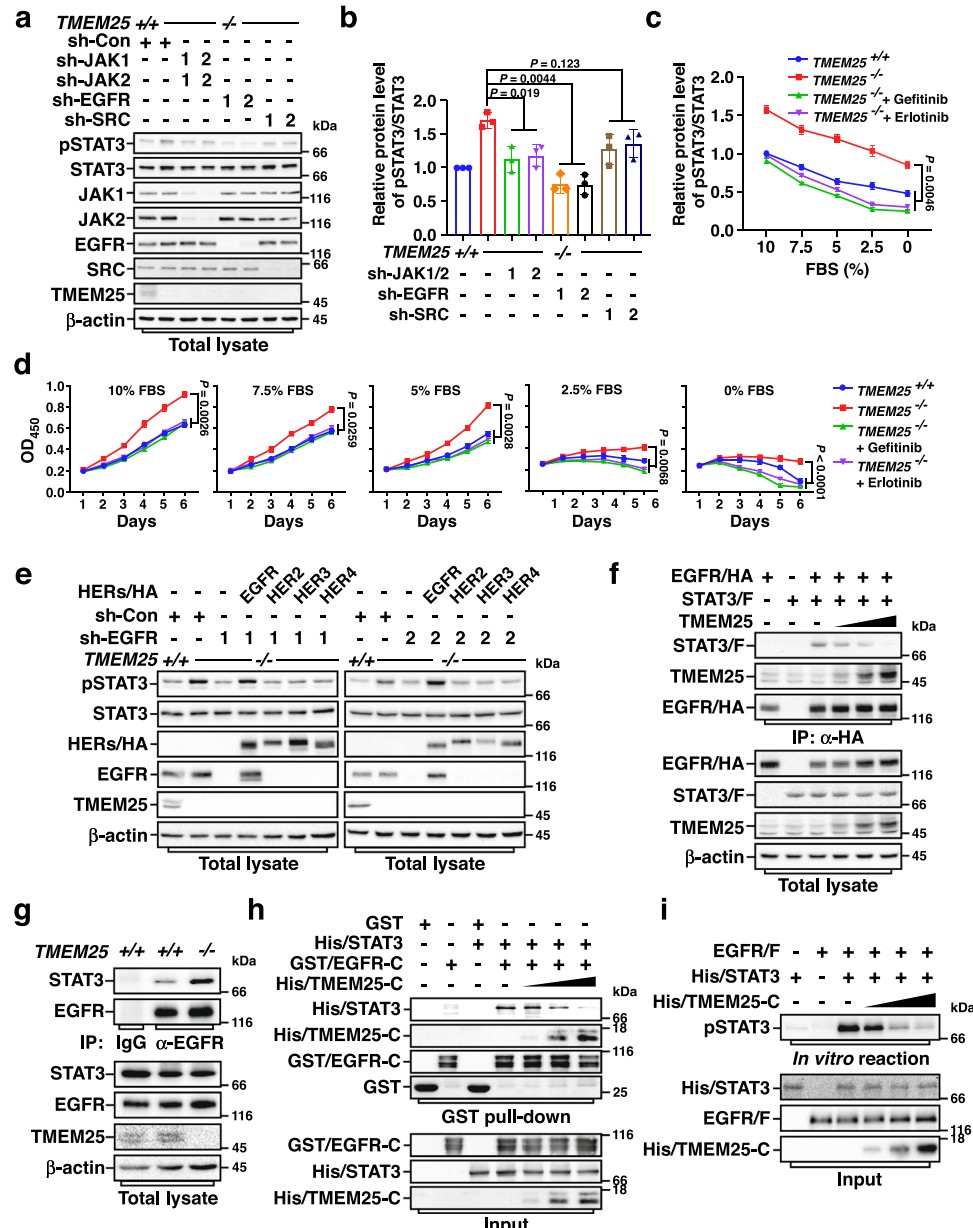

**Fig. 4 | TMEM25 inhibits EGFR-mediated STAT3 phosphorylation. a, b** EGFR is required for *TMEM25* knockout-induced STAT3 phosphorylation. Wild-type or *TMEM25*[−/−] MDA-MB-231 cells transduced with lentivirus encoding control shRNA (sh-Con), shRNAs against JAK1 (sh-JAK1-1/2), JAK2 (sh-JAK2-1/2), EGFR (sh-EGFR-1/2), or SRC (sh-SRC-1/2) as indicated were serum starved overnight before subjected to immunoblotting assay (**a**). Relative pSTAT3/STAT3 ratios were determined and plotted as mean ± SEM (*n* = 3 individual experiments) (**b**). *P* values were determined by one-way ANOVA followed by Tukey test. **c** *TMEM25* knockout enhances STAT3 phosphorylation via EGFR in different serum conditions. Wild-type or *TMEM25*[−/−] MDA-MB-231 cells grown in indicated FBS concentrations were treated with/without EGFR inhibitor Gefitinib (20 μM, 24 h) or Erlotinib (20 μM, 4 h) before subjected to immunoblotting assay. Relative pSTAT3/STAT3 ratios were determined and plotted as mean ± SEM (*n* = 3 individual experiments). *P* values were determined by one-way ANOVA followed by Tukey test. **d** *TMEM25* knockout-promoted cell proliferation and survival requires EGFR activity. Cell viabilities of wild-type and *TMEM25*[−/−] MDA-MB-231 cells cultured in indicated concentrations of FBS with/without Gefitinib (20 μM) or Erlotinib (20 μM) were

determined and plotted as mean ± SEM (*n* = 3 individual experiments). *P* values were determined by one-way ANOVA followed by Tukey test. **e** EGFR induces STAT3 phosphorylation in *TMEM25*[−/−] cells. *TMEM25*[−/−] MDA-MB-231 cells with indicated combinations of sh-Con or sh-EGFR-1/2 and EGFR/HA, HER2/HA, HER3/HA, or HER4/HA were subjected to immunoblotting assay. **f** TMEM25 blocks EGFR and STAT3 interaction. HEK293T cells transfected with indicated combinations of TMEM25, EGFR/HA, or STAT3/F were subjected to Co-IP assay. **g** *TMEM25* knockout enhances EGFR and STAT3 interaction. *TMEM25*[+/+] or *TMEM25*[−/−] MDA-MB-231 cells were subjected to Co-IP assay. **h** TMEM25 inhibits STAT3 and EGFR interaction in vitro. Bacterially produced His-tagged STAT3 (His/STAT3), His-tagged TMEM25 cytosolic domain (His/TMEM25-C), and GST-tagged EGFR cytosolic domain (GST/EGFR-C) were subjected to GST pull-down assay. **i** TMEM25 blocks EGFR-mediated STAT3 phosphorylation in vitro. Affinity-purified EGFR/F from HEK293T and bacterially produced His/STAT3 and His/TMEM25-C were subjected to in vitro kinase assay. The blotting experiments are representative of at least three biologically independent replicates (**a**, **e**–**j**). Source data are provided as a Source Data file.

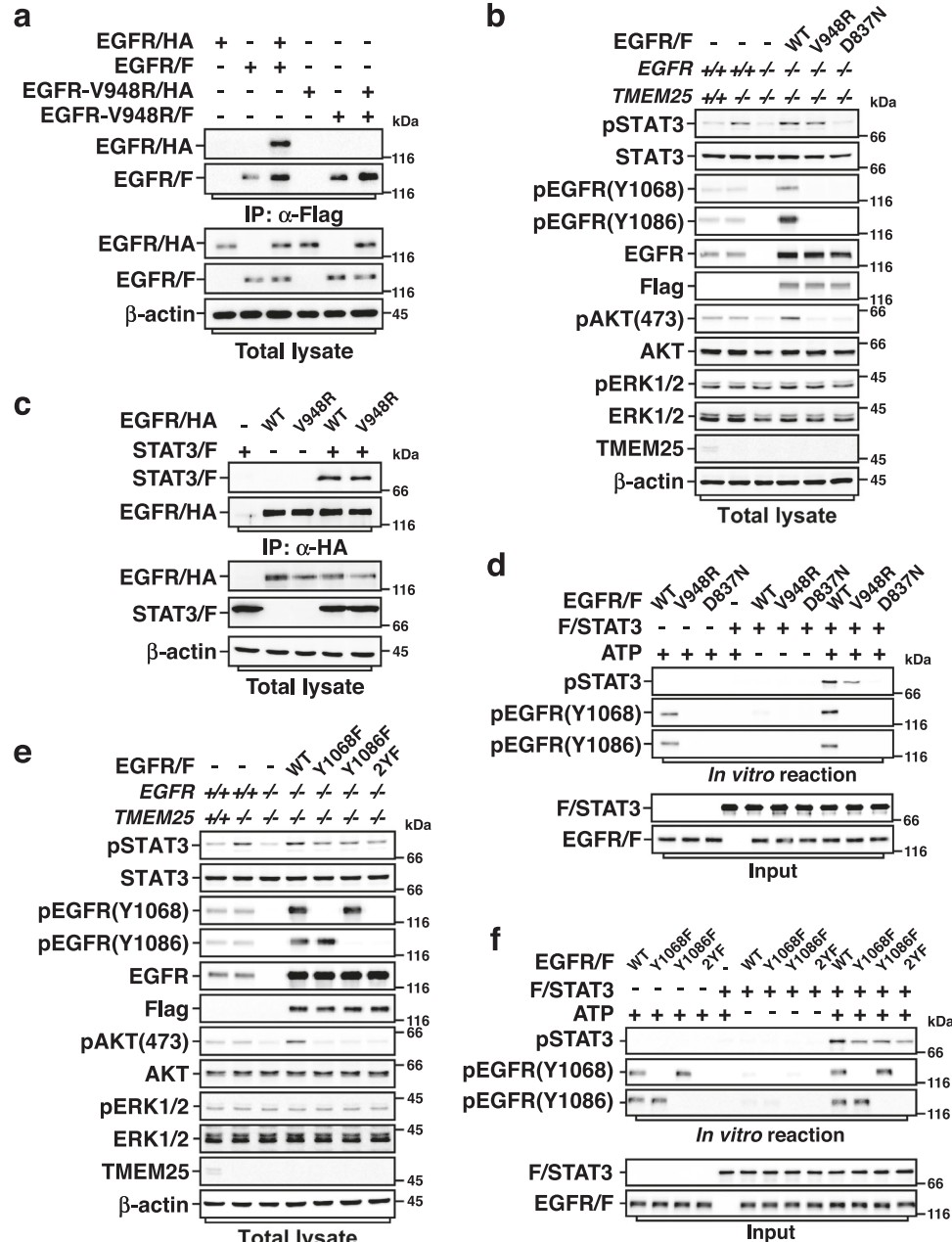

**Fig. 5 | Monomeric EGFR is able to phosphorylate STAT3 in the absence of TMEM25. a** EGFR-V948R mutant does not form dimer. HEK293T cells transfected with indicated combinations of EGFR/HA, EGFR/F, EGFR-V948R/HA, or EGFR-V948R/F were subjected to Co-IP assay. **b** EGFR-V948R is able to phosphorylate STAT3. Lentivirus encoding Flag-tagged wild-type (WT) EGFR, EGFR-V948R, or EGFR-D837N were transduced into *TMEM25* and *EGFR* double knockout (*TMEM25⁻ᐟ⁻EGFR⁻ᐟ⁻*) MDA-MB-231 cells as indicated. The cells were serum starved overnight and then subjected to immunoblotting assay. **c** EGFR-V948R binds STAT3. HEK293T cells transfected with indicated combinations of STAT3/F and EGFR/HA (WT or V948R) were subjected to Co-IP assay. **d** EGFR-V948R can directly phosphorylate STAT3 in vitro. Exogenously expressed EGFR/F (WT, V948R, or D837N) in *EGFR⁻ᐟ⁻* MDA-MB-231 cells were serum starved overnight and then affinity purified for subjecting to in vitro

kinase assay with bacterially produced STAT3. **e** Phosphorylation of Y1068 or Y1086 is not essential for EGFR-mediated phosphorylation of STAT3 in the absence of TMEM25. Lentivirus encoding Flag-tagged wild-type EGFR, EGFR-Y1068F, EGFR-Y1086F, or EGFR-Y1068,1086F (2YF) were transduced into *TMEM25⁻ᐟ⁻EGFR⁻ᐟ⁻* MDA-MB-231 cells as indicated. The cells were serum starved overnight and then subjected to immunoblotting assay. **f** EGFR-Y1068F, EGFR-Y1086F, and EGFR-2YF can phosphorylate STAT3 in vitro. Exogenously expressed EGFR/F (WT, Y1068F, Y1086F, or 2YF) in *EGFR⁻ᐟ⁻* MDA-MB-231 cells were serum starved overnight and then affinity purified for subjecting to in vitro kinase assay with bacterially produced STAT3. The blotting experiments are representative of at least three biologically independent replicates (**a–f**). Source data are provided as a Source Data file.

substantially increased tyrosine phosphorylation of wild-type EGFR. However, no matter with or without EGF treatment, no tyrosine phosphorylation of EGFR-V948R could be detected (Supplementary Fig. 8d). We confirmed that most of EGFR are monomers under serum starvation using crosslinker disuccinimidyl suberate (DSS)

(Supplementary Fig. 8e, f). Hence, these results indicated that monomeric EGFR could phosphorylate STAT3 in the absence of TMEM25 in a tyrosine autophosphorylation-independent manner, giving a good explanation why STAT3 in *TMEM25⁻ᐟ⁻* cells was heavily phosphorylated even without EGF treatment (Fig. 3a and Supplementary Fig. 5a).

## TMEM25 in human TNBC and its potential therapeutic application

To explore the clinical relevance of TMEM25 in human TNBC, we examined the TMEM25 protein levels and phospho-STAT3 levels in 28 human TNBC specimens by immunoblotting assay. In good agreement with the decrease of mRNA levels of TMEM25 in TNBC compared with normal mammary tissues (Supplementary Fig. 2c), we found that TMEM25 protein levels were markedly decreased in 23 out of 28 TNBC samples compared with that in the matched adjacent normal tissues. Meanwhile, among the 23 TMEM25 decreased TNBC samples, 19 had increased phospho-STAT3 levels compared with that in the matched adjacent normal tissues, presenting a negative correlation between decrease of TMEM25 and increase of phospho-STAT3 in TNBC samples (Fig. 6a and Supplementary Fig. 9).

Next, we attempted to test the potential application of TMEM25 for therapeutic purpose. For this end, we injected adeno-associated virus (AAV) encoding wild-type TMEM25 or two TMEM25 mutants (R326W and L338F), which were found in human colorectal cancers but not in human breast cancers in the catalogue of somatic mutations in cancer (COSMIC) database (https://cancer.sanger.ac.uk/cosmic), into mammary fat pad of 8-week-old female *MMTV-PyMT* transgenic mice. TMEM25-R326W and TMEM25-L338F could not effectively block the interaction between EGFR and STAT3, therefore could not inhibit the phosphorylation of STAT3 as wild-type TMEM25 did (Supplementary Fig. 10a, b).

Strikingly, injection with wild-type TMEM25, but not TMEM25-R326W or TMEM25-L338F, encoded AAV showed significant inhibitory effect on the spontaneous TNBC tumor growth in *MMTV-PyMT* transgenic mice (Fig. 6b–d). We confirmed that the phosphorylation levels of STAT3 in primary tumor tissues from mice injected with wild-type TMEM25-encoded AAV were significantly lower than those from mice injected with control AAV, or TMEM25-R326W or TMEM25-L338F encoded AAV by immunoblotting assay (Supplementary Fig. 10c). In addition, immunohistochemistry assay also demonstrated that protein levels of Ki67 and phosphorylated STAT3 in the tumors were inhibited only by injection with AAV encoding wild-type TMEM25, but not TMEM25-R326W or TMEM25-L338F (Supplementary Fig. 10d). Accordingly, supplying wild-type TMEM25, but not TMEM25-R326W or TMEM25-L338F, by AAV significantly prolonged the overall survival time of *MMTV-PyMT* mice (Fig. 6e).

We next further examined the effect of providing wild-type TMEM25 in a patient-derived xenograft (PDX) TNBC tumor model, and found that supply of TMEM25 by AAV could also significantly inhibit PDX TNBC tumor growth (Fig. 6f–h). The inhibitory effect of TMEM25 on phosphorylation of STAT3 in PDX tumors were confirmed by immunoblotting assay (Supplementary Fig. 10e). Meanwhile, decreases of Ki67 and phosphorylated STAT3 in PDX tumors injected with AAV encoding TMEM25 were also confirmed by immunohistochemistry assay (Supplementary Fig. 10f). Hence, our study revealed an underlying mechanism for preventing aberrant activation of EGFR/STAT3 signaling by TMEM25 (Fig. 6i), and demonstrated a potential usage of TMEM25 in targeted therapy for human TNBC.

## Discussion

RTKs are a large family of cell-surface receptors sharing a similar structural architecture composed of an extracellular domain for ligand binding, a single-pass transmembrane domain, and a cytosolic tyrosine kinase domain. In general, it requires ligand-induced formation of dimer/oligomer to activate RTKs. Although some RTKs such as insulin receptor may exist as dimers or oligomers without presence of ligand, binding of ligand is still required to switch their conformation from inactive states to active states[39, 40]. A recent study demonstrated that monomeric tropomyosin-related tyrosine kinase B (TrkB) can bind its ligand brain-derived neurotrophic factor (BDNF) to induce its autophosphorylation and downstream ERK1/2 signaling,

presenting an example of monomeric signaling mode for RTK[41]. Nevertheless, despite of all the different activating mechanisms, ligand-induced conformational change is currently believed necessary for activation of all wild-type RTKs. Our study, however, demonstrated that EGFR monomer can directly bind and phosphorylate STAT3 to initiate downstream signaling independent of ligand binding, presenting a ligand-independent monomeric RTK-triggered signaling mode that may lead to a further understanding of RTK activation mechanisms. This also brings forward a concern that targeting dimerization of RTKs might not always be a suitable therapeutic approach because some of them may also trigger downstream signaling by monomers in certain physiological or pathological contexts.

Given the fact that EGFR is frequently overexpressed and TMEM25 expression is frequently decreased in TNBC, our study suggests that EGFR monomer might be a major instigator to elicit STAT3 signaling in TNBC cells, especially when growing in vivo with limited supply of growth factors. Our study also well explains why targeting EGFR dimerization using cetuximab, an EGFR monoclonal antibody that binds to domain III of EGFR to block ligand binding and meanwhile to prevent EGFR from adopting the extended conformation required for dimerization[42], showed no effect on improving outcome of TNBC patients[10–12]. Because autophosphorylation of EGFR is not essential for EGFR monomer to phosphorylate STAT3, we speculate that binding of STAT3 induces a conformational change in the EGFR kinase domain. This is in agreement with previous studies showing that activation of EGFR family members requires allosteric regulation rather than phosphorylation of the activation loop[32].

It is remarkable that monomeric EGFR specifically activates STAT3 but not STAT5. Interestingly, although both STAT3 and STAT5 are frequently activated in various types of cancer, some studies indicate that STAT3 plays a more potent role in promoting tumor development[14, 15, 29]. Notably, STAT3 and STAT5 may play distinct or even opposite roles in affecting the progression of breast cancer[19]. It is clear that constitutive activation of STAT3 is most often associated with TNBC and leads to malignancy of cancer cells; activation of STAT5, however, is often associated with other subtypes of breast cancer and is considered acts as a favorable marker[43]. Co-activation of STAT3 and STAT5 shows a better prognosis than activation of STAT3 alone[19, 43, 44]. Therefore, it is conceivable that the monomeric EGFR-mediated STAT3 activation might play a key role in promoting malignancy of TNBC.

Loss-of-function mutations of tumor suppressor genes and gain-of-function mutations of proto-oncogenes are frequently involved in tumor development and progression. Although we did not find clinical mutation of TMEM25 affecting its checkpoint function on monomeric-EGFR/STAT3 signaling in breast cancer, we found quite a few clinical mutations of TMEM25 resulting in a loss of this checkpoint function in other types of cancer including colorectal cancer. These mutations are located in the cytosolic domain of TMEM25, which may induce conformational changes of TMEM25 cytosolic domain to affect its binding to the cytosolic domain of EGFR, therefore allowing STAT3 to access to EGFR. Hence, different types of cancer may overcome the TMEM25 obstacle via different mechanisms, e.g. decrease of the expression level and loss-of-function mutations, and this might be the reason why a mere comparison of TMEM25 expression levels did not show a significant difference of the overall survival rates in colorectal cancer patients (Supplementary Fig. 2b). Therefore, it will be necessary to assess the correlation between TMEM25 and overall survival rates of cancer patients by distinguishing these mutations from expression level changes. Furthermore, it will be of interests to investigate whether there is any mutation in EGFR that impedes the interaction between TMEM25 and EGFR to trigger the activation of STAT3.

Comparing with other types of breast cancer, TNBC shows more aggressive clinical behavior and poorer prognosis. Endocrine therapy

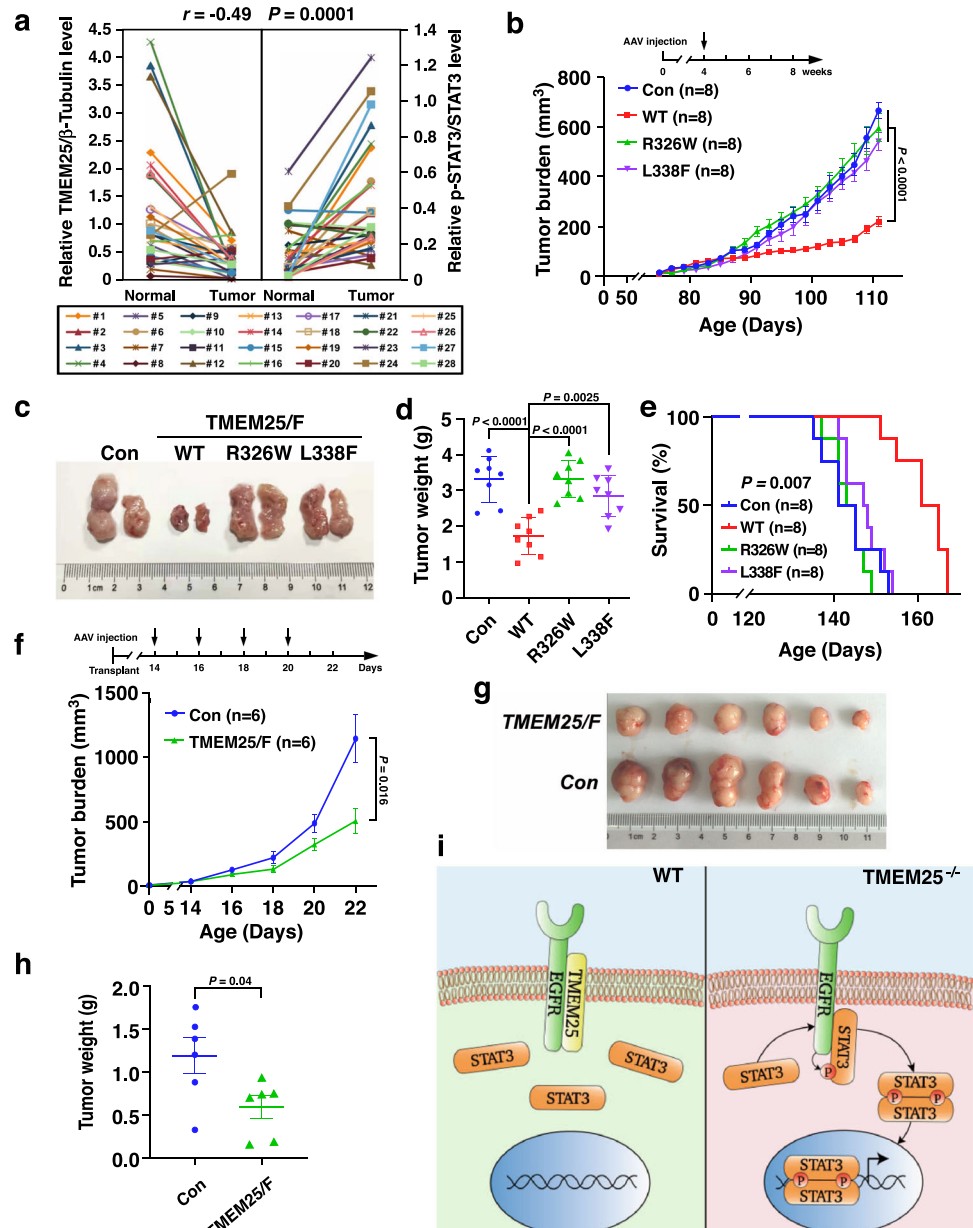

**Fig. 6 | TMEM25 expression and mutations in clinical samples and potential therapeutic application of TMEM25 in TNBC treatment. a** TMEM25 protein levels are frequently decreased and negatively correlated with phospho-STAT3 levels in TNBC. TMEM25 and phospho-STAT3 levels in 28 human TNBC samples were determined and their correlation coefficient *r* value and *p* value were determined by Spearman's rank correlation analysis and two-tailed unpaired *t* test. **b–d** Wild-type but not R326W or L338F TMEM25 suppresses tumor growth in *MMTV-PyMT* mice. *MMTV-PyMT* mice were orthotopically injected into mammary fat pad once at 8 weeks old with control AAV, or AAV encoding TMEM25 (WT, R326W, or L338F). Tumor volumes were measured and plotted as mean ± SEM (*n* = 8 animals per group). *P* values were determined by two-way ANOVA followed by Tukey test (**b**). Representative tumors obtained from one mouse injected with control AAV or AAV encoding indicated TMEM25 were presented (**c**). All tumors were weighted and plotted as mean ± SEM (*n* = 8 animals per group). *P* values were determined by one-way ANOVA followed by Tukey test (**d**). **e** Supply of wild-type

but not R326W or L338F TMEM25 prolongs survival time of *MMTV-PyMT* mice. Overall survival rates of *MMTV-PyMT* mice injected with control AAV or AAV encoding indicated TMEM25 were determined (*n* = 8 animals per group). *P* values were determined by log-rank (Mantel-Cox) test, 95% confidence interval of ratio. **f–h** AAV-delivered TMEM25 suppresses tumor growth in patient-derived xenograft (PDX) models. NCG mice (6–8 weeks old) were intratumorously injected with control AAV, or AAV encoding TMEM25 every other day. Tumor volumes were measured and plotted as mean ± SEM (*n* = 6 animals per group). *P* values were determined by two-way ANOVA followed by Tukey test (**f**). Tumors obtained from NCG mice 22 days after transplantation and injected with control AAV or AAV encoding TMEM25 were presented (**g**). All tumors were weighted and plotted as mean ± SEM (*n* = 6 animals per group). *P* values were determined by two-tailed unpaired Student's *t* test (**h**). **i** A schematic model of TMEM25/EGFR/STAT3 signaling. Source data are provided as a Source Data file.

---

by blocking the hormone receptor activities has been proven effective to treat breast cancer with the presence of estrogen receptor (ER) and/or progesterone receptor (PR). Meanwhile, targeting HER2 has shown significant efficacy in the treatment of HER2-positive breast cancer patients. However, no targeted therapy has been approved for

treatment of TNBC patients yet due to its lack of expression of ER, PR, and HER2[45, 46]. In this study, by supplying TMEM25, we successfully suppressed growth of transplant TNBC tumors and spontaneous TNBC tumors, pointing out a potential targeted therapy for TNBC patients. In addition, an alternative strategy to block the monomeric

EGFR-mediated hyperactivation of STAT3 could potentially be achieved by developing short peptides or small molecule compounds to block the interaction between EGFR and STAT3, which may provide lead compounds for TNBC drug development.

## Methods

All mouse experiments were approved by the Institutional Animal Care and Use Committee of Xiamen University and were performed following the guidelines for the use of laboratory animals. The collection and use of clinical samples were in accordance with research ethics board approval from Xiamen University and the Affiliate Hospitals.

### DNA constructs

The complementary DNAs (cDNA) of human TMEM25 was cloned from human embryonic kidney HEK293T cells using reverse transcription polymerase chain reaction (RT-PCR). cDNAs of human STAT3, EGFR, HER2, HER3 and HER4 were gifts from Dr. J. Han. Mutations of TMEM25 and EGFR were generated by PCR–based site-directed mutagenesis. Cloning for protein expression in mammalian cells was carried out using a modified pCMV5 vector for transfection, pBOBI vector for lentivirus infection, and AAV-9 vector for adeno-associated virus infection. pGEX-4T-1 and pProEX were used for bacterial expression of proteins. The lentiviral-based vector pLL3.7 was used for short hairpin RNA (shRNA) expression. The sequences used for human EGFR shRNA-1 and shRNA-2 are 5′-GCTGAGAATGTGGAATACCTA-3′ and 5′-GCTGCTCTGAAATCTCCTTT-3′, respectively; for human EGFR gRNA-1 and gRNA-2 are 5′-ATAACTGTGAGGTGGTCCTT-3′ and 5′-AATTCGCTCCACTGTGTTGA-3′, respectively; for human TMEM25 shRNA-1 and shRNA-2 are 5′-CCGTCCAACCTTCAGCTCA-3′ and 5′-CGGCAGATGGCTCAGAACA-3′, respectively; for human JAK1 shRNA-1 and shRNA-2 are 5′-GCACTCCTCCTTGTGGAAAGA-3′ and 5′-GCTGCCAGCTGATCTGAAATG-3′, respectively; for human JAK2 shRNA-1 and shRNA-2 are 5′-GCTTTGTCTTTCGTGTCATTA-3′ and 5′-GGCTTCCCGGCTGCCCGAAGT-3′, respectively; for human SRC shRNA-1 and shRNA-2 are 5′-GCTGAGAATGTGGAATACCTA-3′ and 5′-GCTGCTCTGAAATCTCCTTT-3′, respectively; for human STAT3 shRNA-1 and shRNA-2 are 5′-ACAATCTACGAAGAATCAA-3′ and 5′-GCAACAGATTGCCTGCATTGG-3′, respectively. The scramble sequence 5′-TTCTCCGAACGTGGCACGA-3′ was used for a control shRNA.

### Antibodies and chemical reagents

Mouse anti-β-actin (1:2000, sc-47778), anti-GST (1:2000, sc-138), anti-GFP (1:2000 for Western Blot (WB), 1:200 for immunoprecipitation (IP), sc-9996) were purchased from Santa Cruz Biotechnology (Santa Cruz, CA, USA); mouse anti-FLAG (M2) (1:2000 for WB, 1:200 for IP, F1804), anti-β-tubulin (1:2000, T4026), and rabbit anti-TMEM25 (1:1000 for WB, 1:100 for IP, HPA012163) from Sigma-Aldrich (St Louis, MO, USA); rabbit anti-Ki67 (1:200 for immunohistochemistry (IHC), 12202T), anti-phospho-STAT3 (1:2000 for WB, 1:100 for IHC, Y705) (9145L), anti-STAT3 (1:2000, 12640), anti-phospho-EGFR (Y1068) (1:2000, 3777), anti-phospho-EGFR (1:2000, Y1086) (2220), anti-EGFR (1:2000 for WB, 1:100 for IP, 1:100 for immunofluorescence (IF), 4267), anti-HER3 (1:2000, 12708), anti-HER4 (1:2000, 4795), anti-phospho-STAT5 (Y694) (1:2000, 9351), anti-STAT5 (1:2000, 94205), anti-phospho-AKT (S473) (1:2000, 9271), anti-AKT (pan) (1:2000, 4691), anti-phospho-ERK1/2 (1:2000, T202/Y204) (4370), anti-ERK1/2 (1:2000, 9102), anti-JAK1 (1:2000, 3344), anti-JAK2 (1:2000, 3230), and anti-SRC (1:2000, 2109) antibodies from Cell Signaling Technology; rabbit anti-HER2 (1:2000, 18299-1-AP) from Proteintech; rat anti-HA antibody (1:2000, 11867431001) and mouse anti-HA antibody (1:200 for IP, 11666606001) from Roche; HRP anti-phosphotyrosine (pY20) (1:2000, ab16389) were purchased from Abcam. Alexa Fluor 555 donkey anti-rabbit (1:500, A31572), goat anti-rabbit IgG (H + L) secondary antibody, HRP (1:5000, 31460), goat anti-mouse IgG (H + L) secondary

antibody, HRP (1:5000, 31430), goat anti-rat IgG (H + L) secondary antibody, HRP (1:5000, 31470) were purchased from Thermo Fisher Scientific. EGF (GMP-10605-HNAE) was from Sino Biological. Cell Counting Kit-8 (CCK-8), JAK1/2 inhibitors Ruxolitinib (HY-50856) and Baricitinib (HY-15315), SRC inhibitors Bosutinib (HY-10158) and Saracatinib (HY-10234), EGFR inhibitors Gefitinib (HY-50895) and Erlotinib (HY-50896), STAT3 inhibitor NSC74859 (HY-15146), and disuccinimidyl suberate (DSS) (HY-W019543) were from MedChemExpress. Collagenase IV (C5138) was from Sigma-Aldrich.

### Cell culture, transfection, and lentivirus infection

Human embryonic kidney HEK293T (CRL-3216), mouse breast cancer 4T1 (CRL-2539), human breast cancer MDA-MB-231 (HTB-26), MCF7 (HTB-22), BT549 (HTB-122), and HCC1937 (CRL-2336) cell lines were obtained from ATCC. HEK293T, MDA-MB-231, and MCF7 cells were cultured in high-glucose Dulbecco's modified Eagle's medium (DMEM); BT549, HCC1937 and 4T1 cells were cultured in RPMI 1640, all supplemented with 10% (v/v) fetal bovine serum (FBS) (PAN) and streptomycin and penicillin (100 U/ml; HyClone) at 37 °C in a humidified 5% $CO_2$ incubator. The cell lines were routinely tested and found negative for mycoplasma. For EGF treatment, cells were washed with phosphate-buffered saline (PBS), cultured in DMEM without FBS overnight, and then treated with EGF for indicated times. Transient transfection was performed using the poly-ethylenimine (PEI) method. Plasmids and PEI mixture with a ratio of 3:1 (w/w) were added to cell culture, and it was replaced with fresh medium 6 h after transfection. Recombinant lentivirus for infection was generated using the ViraPower Lentiviral Expression System (Invitrogen).

### Treatment of EGF and inhibitors of EGFR, JAK1/2, SRC or STAT3

For EGF treatment, cells were serum starved overnight and then treated with 100 ng/ml EGF[47]. Treatment of EGFR inhibitors Gefitinib (20 μM, 24 h)[48] and Erlotinib (20 μM, 4 h)[49], JAK1/2 inhibitors Ruxolitinib (0.5 μM, 1 h)[50] and Baricitinib (2.5 μM, 24 h)[51], SRC inhibitors Bosutinib (10 μM, 1 h)[52] and SRC inhibitor Saracatinib (10 μM, 24 h)[53], and STAT3 inhibitor NSC74859 (50 μM)[54] were carried as previously reported.

### Generation of primary *TMEM25*^+/+ and *TMEM25*^−/− MEF cells

*TMEM25*^+/+ and *TMEM25*^−/− MEF cells were generated using E13.5 embryos FVB *TMEM25*^+/+ and *TMEM25*^−/− mice. Embryos were minced with razor blades after removal of heads, limbs and visceral tissues, and then trypsinized (0.25% Trypsin-EDTA) for 15 min at 37 °C. The cells were resuspended in DMEM with 10% FBS, and then seeded in 6 cm plate to culture at 37 °C in a humidified 5% $CO_2$ incubator.

### Generation of *TMEM25*^−/− and *EGFR*^−/− cell lines

The guide RNA sequences targeting *TMEM25* were designed using the CRISPR design tool (https://zlab.bio/guide-design-resources). The two gRNA sequences for human *TMEM25* −1 and −2 are 5′-CCACGCCTT CACCTGCCGGG-3′ and 5′-TCCAGGTGACATTGGCCGGC-3′. The two gRNA sequences for mouse *TMEM25* −1 and −2 are 5′-CTTGGCACAC AACCTCTCGG-3′ and 5′-TCCAGGTACCAGGCTAATCG-3′. The two gRNA sequences for human *EGFR* −1 and −2 are 5′-ATAACTGTG AGGTGGTCCTT-3′ and 5′-AATTCGCTCCACTGTGTTGA-3′. pLenti-CRISPR V2 viral vector was used to express sgRNAs in cells. The pool of *TMEM25*^−/− cells were selected by treatment with puromycin (2 μg/ml) for 7 days. The efficiency of *TMEM25* and *EGFR* knockout was assessed by quantitative reverse transcription-PCR (qRT-PCR) and immunoblotting assays.

### Treatment of tumor tissues with collagenase

Mammary tumor tissues from mice were minced using a scalpel and then incubated in 5 ml HBSS supplemented with 1.5% (w/v) collagenase

IV (Sigma-Aldrich) at 37 °C for 2 h with gentle agitation. Cells were collected by centrifuging at 230 × g for 5 min. The cells were resuspended in HBSS and then filtered using 70-µm strainer before applied for further examination.

### Cell viability and colony formation assays

Cell growth rate was determined by counting the cell numbers daily after seeding into 96-well plate (triplicate/sample) using a CCK-8 kit following the manufacturer's instruction. Briefly, 10 µl CCK-8 solution was added to each well, incubated for 1 h, and then subjected to measurement of the absorbance at 450 nm using a microplate reader (Tecan Spark).

For colony formation assay, cells were seeded at 24-well plate in 500 µl of 0.25% (w/v) Noble agar (BD) with complete medium. The plates were precoated with 500 µl of 0.5% (w/v) Noble agar, and 300 µl of overlay medium was added after cell plating. The cells were cultured at 37 °C in a humidified 5% $CO_2$ incubator for 1–3 weeks. The overlay medium was changed every 3 days. Colonies formed in soft agar were photographed using a Nikon eclipse Ti-U inverted microscope and NIS-Elements software.

### Transplant tumor formation assay

For generation of transplant tumors, 4T1 cells ($5 \times 10^5$ cells suspended in 15 µl PBS) or MDA-MB-231 cells ($2 \times 10^6$ cells suspended in 100 µl PBS) were orthotopically injected into the mammary fat pads of female BALB/c mice (6 weeks old) or nude mice, respectively. Volumes of tumor were measured with digital calipers every other day for about 2 weeks after the tumors were visible. Female BALB/c mice and nude mice were purchased from and housed in the Laboratory Animal Center of Xiamen University (China). All mice were housed under specific pathogen-free (SPF) conditions with 12 h light/12 h dark cycle at 21–24 °C, and humidity at 40–60%, and were fed with a standard chow diet at the Xiamen University Laboratory Animal Center. All procedures involving mice for transplant tumor growth assays were approved by the Institutional Animal Care and Use Committee of Xiamen University. At the end of animal studies, all mice were euthanized by inhaling carbon dioxide.

### Generation of TMEM25$^{-/-}$, TMEM25$^{wt/tg}$, TMEM25$^{-/-}$-MMTV-PyMT, and TMEM25$^{wt/tg}$-MMTV-PyMT mice

C57BL/6 background *TMEM25$^{fl/fl}$* mice were crossed with *EIIA* cre to generate whole body *TMEM25$^{+/-}$* mice. The resulting pups were backcrossed to C57BL/6 mice to remove the *EIIA-Cre*. C57BL/6 *TMEM25$^{+/-}$* mice were crossed with FVB mice for 10 generations to generate *TMEM25$^{+/-}$* mice on the FVB background. *TMEM25$^{+/-}$-MMTV-PyMT* mice were generated by crossing female FVB *TMEM25$^{+/-}$* mice with male FVB *MMTV-PyMT* transgenic mice. Male FVB *TMEM25$^{+/-}$-MMTV-PyMT* and female FVB *TMEM25$^{+/-}$* mice were used to generate *TMEM25$^{+/+}$-MMTV-PyMT* and *TMEM25$^{-/-}$-MMTV-PyMT* littermates for experiments. FVB background *TMEM25$^{wt/tg}$* mice were ordered from Shanghai Model Organisms Center, Inc. Female FVB *TMEM25$^{wt/tg}$* mice were crossed with male FVB *MMTV-PyMT* mice to generate FVB *TMEM25$^{wt/tg}$-MMTV-PyMT*. Male FVB *TMEM25$^{wt/tg}$-MMTV-PyMT* and female FVB *TMEM25$^{+/+}$* mice were used to generate *TMEM25$^{wt/wt}$-MMTV-PyMT* and *TMEM25$^{wt/tg}$-MMTV-PyMT* littermates for experiments. All mice were housed under specific pathogen-free (SPF) conditions with 12 h light/12 h dark cycle at 21–24 °C, and humidity at 40–60%, and were fed with a standard chow diet at the Xiamen University Laboratory Animal Center. All procedures involving mice were approved by the Institutional Animal Care and Use Committee of Xiamen University.

### Measurement of tumor size and survival percentage

Volumes of tumor were measured with digital calipers after the tumors were palpable. Mice were sacrificed before the total tumor burden reached 2 $cm^3$. For survival measurement, mice were sacrificed when tumors reached 25 mm in diameter or ulcerated according to the Xiamen University Laboratory Animal Center-approved endpoints. At the end of animal studies, all mice were euthanized by inhaling carbon dioxide.

### Orthotopically injection of AAV

Adeno-associated virus (AAV) vector (alone or cloned with target gene) was transfected into HEK293T cells together with AAV helper plasmid 2/9, and △T6 plasmid (1:1:3) for AAV packaging. AAV was collected from both media and cell lysate (lysed by 5–6 cycles of freeze and thaw) by adding 1/4 volume of 40% PEG8000, 2.5 M NaCl, incubated overnight at 4 °C and then subjected ultracentrifugation at 350,000×g for 75 min using Iodixanol gradients. Collected virus was then transferred to 100 K columns to remove Iodixanol and to be purified and concentrated. The purified virus was orthotopically injected into multi-sites (10 µl/site) of mammary fat pad in mice.

### Immunoprecipitation, immunoblotting, and GST pull-down assays

Cells were lysed on ice with lysis buffer TNTE 0.5% (50 mM Tris-HCl, pH 7.5, 150 mM NaCl, 1 mM EDTA, and 0.5% Triton X-100, containing 10 µg/ml pepstatin A, 10 µg/ml leupeptin, and 1 mM PMSF). Lysates were separated by SDS-PAGE and transferred to polyvinylidene fluoride membranes and then incubated with the indicated antibodies for immunoblotting assay. For immunoprecipitation, cells grown in 100-mm dish were lysed in 1 ml lysis buffer and cleared by centrifuging at 20,000×g (10 min, 4 °C). A 50 µl aliquot of the lysate was taken for IB assay to examine protein expression and the remaining lysate was added 1 µg appropriate antibody and 30 µl protein G agarose beads and incubated at 4 °C for 3 h before washed 5 times with TNTE buffer. The beads were then boiled in 20 µl 2×SDS loading buffer for 5 min to collect the samples for SDS-PAGE. For GST pull-down assay, bacterially expressed GST-tagged EGFR cytosolic domain (GST/EGFR-C) was purified using glutathione sepharose beads in TNTE 0.5% buffer, and His-tagged STAT3 (His/STAT3) and TMEM25 cytosolic domain (His/TMEM25-C) were purified using $Ni^{2+}$-NTA-agarose chromatography.

### EGFR dimerization assay

Overnight serum-starved MDA-MB-231 cells were treated 5 min with EGF (100 ng/ml) at 37 °C and then incubated with 1 mM crosslinker DSS t at 4 °C for 1 h. The crosslinking reaction was terminated by incubating with 20 mM Tris-HCl (pH 7.5) for 15 min. The cells were then washed with cold PBS immediately and harvested for immunoblotting analysis.

### In vitro kinase assays

Flag-tagged EGFR (WT or V948R) was affinity purified from *EGFR$^{-/-}$* MDA-MB-231 cells and incubated with bacterially produced STAT3 and TMEM25-C in kinase reaction buffer (20 mM Tris-HCl pH 7.5, 10 mM $MgCl_2$, 1 mM DTT, 25 µM ATP) in a total volume of 20 µl at 37 °C for 1 h before subjected to immunoblotting assay.

### Immunofluorescence and immunohistochemistry assays

MDA-MB-231 cells stably expressing TMEM25/GFP were seeded on glass coverslips. Cells were washed three times with PBS, fixed with 4% paraformaldehyde, and permeabilized with 0.25% Triton X-100. EGFR were detected using rabbit anti-EGFR antibody followed by Alexa Fluor 555-conjugated secondary antibody. Images were acquired by using a Zeiss LSM 780 laser-scanning confocal microscope and ZEN 2010 software (Carl Zeiss).

For immunohistochemistry assay, tumors and lung tissues were fixed in 4% (v/v) paraformaldehyde (PFA), embedded in paraffin, sectioned at 3 µm, and then stained with haematoxylin and eosin (H&E) following standard procedures. Immunohistochemistry was

performed by using the UltraSensitiveTM SP kit (MXB, KIT-9720) with appropriate antibodies. Chromogenic revelation was performed with DAB kit (MXB, DAB-1031). Images were obtained using Leica Aperio Versa 200 and Leica DM4B.

## Quantitative real-time PCR

Total RNAs extracted from tissues or cells using TRIzol were applied to synthesize cDNAs with ReverTra Ace qPCR RT kit (Toyobo). Real-time quantitative PCR was performed using SYBR Green PCR Mix (Roche) according to the manufacturer's protocol on a BIO RAD CFX manager. The relative changes of gene expression were determined using the $2^{-\triangle\triangle Ct}$ method and normalized to the internal control GAPDH. The primers for mouse TMEM25 are 5′-ATGGAATTGCCTCTAAGCCAAG-3′ (forward) and 5′-GTACCAGGCTAATCGGGGAGT-3′ (reverse); for mouse GAPDH are 5′-AGGTCGGTGTGAACGGATTTG-3′ (forward) and 5′-TGTAGACCATGTAGTTGAGGTCA-3′ (reverse); for human TMEM25 are 5′-ACCAGCACCTTCACTGTCAC-3′ (forward) and 5′-TGAGCTT CCTGGTACTTGGC-3′ (reverse); for human EGFR are 5′-AGGCACGAGT AACAAGCTCAC-3′ (forward) and 5′-ATGAGGACATAACCAGCCACC-3′ (reverse); for human CCND1 are 5′-GCTGCGAAGTGGAAACCATC-3′ (forward) and 5′-CCTCCTTCTGCACACATTTGAA-3′ (reverse); for human MMP9 are 5′-GGGACGCAGACATCGTCATC-3′ (forward) and 5′-TCGTCATCGTCGAAATGGGC-3′ (reverse); for human HIF1A are 5′-GAACGTCGAAAAGAAAAGTCTCG-3′ (forward) and 5′-CCTTATCAA-GATGCGAACTCACA-3′ (reverse); for human BCL3 are 5′-AACCTGCC TACACCCCTATAC-3′ (forward) and 5′-CACCACAGCAATATGGAGA GG-3′ (reverse); for human SOCS1 are 5′-CACGCACTTCCGCACATTC-3′ (forward) and 5′-TAAGGGCGAAAAAGCAGTTCC-3′ (reverse); for human SOCS3 are 5′-CCTGCGCCTCAAGACCTTC-3′ (forward) and 5′-GTCACTGCGCTCCAGTAGAA-3′ (reverse).; for human GAPDH are 5′- CATGAGAAGTATGACAACAGCCT-3′ (forward) and 5′-AGTCCTTCC ACGATACCAAAGT-3′ (reverse).

## Patient samples

Primary human breast cancer tissue samples and corresponding adjacent normal tissues were obtained in accordance with research ethics board approval from Xiamen University and the Affiliate Hospitals. Informed consent was obtained from all patients. All samples taken after surgery were stocked in liquid nitrogen for further analysis. The information of patients was summarized in Supplementary Table. 1.

## Statistics and reproducibility

GraphPad Prism 8 software was used to analyze all quantitative data. The data are represented as mean ± SEM calculated using GraphPad. Significance was tested using unpaired two-tailed Student's $t$ test, one-way ANOVA with Tukey test and two-way ANOVA with Tukey test. For immunoblotting assay, protein bands were visualized by Sagecreation MiniChemi system and analyzed with Lane 1D software. $P < 0.05$ was considered a statistically significant difference. All data are representative of at least three independent experiments unless otherwise specified.

## Reporting summary

Further information on research design is available in the Nature Portfolio Reporting Summary linked to this article.

## Data availability

All data supporting the findings of this study are available within the article and its Supplementary Information files. The datasets from the cancer genome atlas (TCGA) (https://www.cancer.gov/ccg/research/genome-sequencing/tcga) were analyzed using the UALCAN platform (http://ualcan.path.uab.edu/analysis.html). UCSC Xena platform (https://xena.ucsc.edu) was used for survival analysis in breast cancer and colorectal cancer patients. Source data are provided with this paper.

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

## Acknowledgements

This work was supported by the National Natural Science Foundation of China (82273037, 31970742, 32070761, 91857107, 82002698), the Natural Science Foundation of Fujian Province (2022J05006), Open Research Fund of State Key Laboratory of Cellular Stress Biology, Xiamen University (SKLCSB2019KF009), and the Natural Science Foundation of Guangdong Province (2019A1515110502). We thank Dr. Feng Ding for helping generating the knockout mice and Hannah G. Wang for graphing the model.

## Author contributions

J.B., Z.W., X.Z., T.Z., N.Q., M.X., W.D., Y.Q., L.K., and J.Z. conducted the experiments and analyzed the data. X.C., Q.L., and K.C. performed molecular biology experiments. L.X. and X.L.C. analyzed the data. Z.O., J.G., L.Z., and C.M. provided TNBC clinical samples. S.G. executed PDX experiment. W.M. helped design the experiments. G.F. and T.-J.Z. contributed reagents and designed the experiments. H.-R.W. designed the experiments and wrote the manuscript.

## Competing interests

The authors declare no competing interests.
