## [Peer Review File · Nature Communications]

TMEM25 inhibits monomeric EGFR-mediated STAT3 activation in basal state to suppress triple-negative breast cancer progressionEditorial Note: Parts of this Peer Review File have been redacted as indicated to remove third-party material where no permission to publish could be obtained.

REVIEWER COMMENTS

Reviewer #1 (Remarks to the Author):

The authors have investigated the mechanistic basis for lack of efficacy of EGFR dimerization-blocking antibodies. They find monomeric EGFR-STAT3 signaling is oncogenic in TNBC and must be overcome therapeutically. Evidence of the tumor suppressive role of TMEM25 from the genetically engineered mouse models is convincing and striking. Biochemical evidence of the physical interaction between TMEM25 and EGFR is robust. The manuscript is exceptionally well written and clear. The mutation studies are flawed and need improvement or need to be removed. From what I can tell, there is no justification or need to study the mutation of TMEM as a mechanism of its inactivation in TNBC. The AAV experiments are exciting (here the mutation is appropriate to use as a 'negative control' experimentally).

1. In the introduction and perhaps even abstract, please state the % of TNBC patients with active EGFR signaling axis.
2. For lung metastasis findings in figure 2 and supplementary figure 4 – can authors demonstrate there is an effect on metastasis independent of an effect on primary tumor growth? Or perhaps metastasis is only affected because primary tumor growth rate is altered. One approach would be to normalize lung metastasis burden to primary tumor burden.
3. Figure 2- what is the EGFR status of MMTV pyMT mice in the tumor cells themselves? Is it expressed at the protein level? What is relative amount of monomeric vs dimeric EGFR? Is EGFR physically interacting with TMEM25 in the tumor cells growing in the mice?
4. In human TNBC, is the typical ratio of monomeric vs dimeric EGFR known? Does this correlate with any disease parameters or outcomes?
5. In the introduction, please more clearly communicate the knowledge gap this study addresses. EGFR is known to activate STATs. Is it understood whether that is monomeric or dimerized EGFR? Clearly state the Y2H screen identified a novel (?) interaction between EGFR and TMEM25.
6. Is the mechanism of physical interaction between EGFR and TMEM25 possible to be disrupted with a antibody or steric inhibitor? It would be good if authors have data to address this. At a minimum this possibility should be discussed.
7. In the TNBC cell lines used, does TMEM25 level correlate with proliferation, tumorigenicity, etc?
8. Please edit this sentence for clarity: "we found that TMEM25 protein levels were markedly decreased in 23 out of 28 samples and showed..." – decreased relative to what? Do you mean expression was low? How was low vs high judged? In this same sentence, it is stated there is negative correlation with p-STAT3, suggesting 'TMEM is required for restraining STAT3 activation to suppress TNBC progression.' The correlation does not in fact suggest this sweeping finding. Please remove that phrase.
9. The TMEM25 mutation studies are inherently flawed.
 - a. In text please report what % (and the raw number of patients) with the G59D TMEM25 mutation. Is this mutation expected to be somatic? Nonsynonymous? SIFT/polyphen etc prediction of deleterious impact on protein function?
 - b. Why was the G59D mutation not examined experimentally, since apparently this is the only one detected in a substantial number of TNBC samples?
 - c. The non-TNBC mutations may be completely irrelevant to TNBC. The expression of these mutations in TNBC is not justified, so the experiment in which they are introduced into 231 or 4t1 cells does not make much sense. Authors need to discretely state this in the results section or remove those experiments altogether.
10. The AAV experiment in which WT TMEM inhibited pyMT tumor growth is exciting. The mutation construct here can be evaluated as a tool to inactivate TMEM but again it must be stated very clearly there is no evidence for this mutation being relevant in the TNBC space.
11. Have authors tested STAT3 inhibition in mice +/-TMEM? If not, please discuss. This would be a good experiment.

Minor comments

1. In text there are a couple instances where it is stated that IHC was done on some tumors and readers are referred to an extended data figure. Authors should also include a statement in the text summarizing the result of the IHC experiment.
2. In the introduction please discuss the status of STAT inhibition in TNBC preclinically and clinically
3. Please cite the PiggyBac approach
4. Methods- please provide catalog numbers and concentrations for EGF and all inhibitors. Please provide supplemental data or citation justifying the choice of dosages used
5. Methods- IP section is too simple and lacks citation. Authors should provide sufficient detail so that readers can accurately repeat the experiment
6. Extended fig 5- figure and legend should indicate what the IHC antibody is recognizing. Is it Ki67 again?
7. Why have authors not done IHC for TMEM25 especially in the patient samples? Is there a suitable antibody for IHC?
8. Extended fig 6- some x axis labels are missing

Reviewer #2 (Remarks to the Author):

This is a generally well-written and fairly comprehensive manuscript outlining the molecular and functional role of TMEM25 in tumor formation, and in negatively regulating the ability of monomeric EGFR to activate STAT3. This is accomplished by physical complexing with EGFR.

I have only a few minor comments:

1. The terms "upregulated" and "downregulated" are used a little imprecisely. please state specifically "downregulated" relative to what? There are data relative to normal mammary epithelium and there are data relative to other tumor types. Please specify which comparison is made where in the paper to clarify for the reader.
2. Given the vagueness of the expression statements, it would be of interest to see the expression pattern in the normal human (or even just mouse) mammary epithelium relative to tumor by immunostaining rather than bulk RNA.
3. Some additional in vivo work, perhaps in PDX, or at least orthotopic transplantation of the manipulated cell lines (subcutaneous is well known to show different results in many cases than the inguinal (#4) mammary fat pad). It is possible that the results would be somewhat different, or perhaps even significantly different.

Reviewer #3 (Remarks to the Author):

The epidermal growth factor receptor (EGFR) has been shown to be overexpressed in the majority of triple negative breast cancers (TNBCs), however, its role in the initiation and progression of these cancers remains elusive. Specific to this manuscript, monoclonal antibodies targeting the EGFR family of kinases have not shown promise in the clinic. The authors of this manuscript provide evidence that the EGFR does not respond to monoclonal antibody inhibition in TNBC because the EGFR is signaling as a ligand-independent monomer and thereby does not require ligand binding or dimerization, the mechanism of monoclonal antibody inhibition. They have further evidence that this is mediated by downregulation of the transmembrane protein TMEM25 which at normal expression levels binds to monomeric EGFR and prevents ligand-independent signaling. The authors continue to characterize the ligand-independent monomeric signaling of EGFR to occur through STAT3 but not through MAPK or AKT. While these results have added a new player, TMEM25, into the EGFR signaling field and have the potential to generate great potential in the treatment of TNBC, there are several key experiments missing from this manuscript.

1) The presence of EGFR monomers and the absence of autocrine ligand production are not biochemically confirmed in this manuscript. The presence of monomers and absence of dimers of the EGFR by a technique such as cross-linking followed by non-reducing gel electrophoresis would help to strengthen this major conclusion. It has been published that MDAMB231 cells produce several EGFR ligands in an autocrine fashion. It would strengthen the ligand-independent conclusion of this paper to demonstrate that they are not being produced or binding to EGFR in your model system.

2) When TMEM25 is knocked out and overexpressed in the key tumor growth experiments in Figure 2, there are no companion figures to demonstrate the changes in expression, either in the cells prior to injection in the mouse or in the subsequent mouse tumors. There is one figure, Extended Figure 3 that shows one example of knock out in the cell line and one in the tumor but it is not clear with which experiment those data associate. I understand that these are CRISPR knockouts but the level of TMEM25 expression endogenously in these cells is almost undetectable. For the knockout to have such a profound biological change in cell and tumor growth is perplexing, particularly when the overexpression had much a higher expression but does not have the same significance with respect to biological impact. This is particularly key with the transgenic models (Extended Figure 4f). There are no detectable differences via Western blot between the knockout and wild-type TMEM25 expression.

3) Only looking at two sites of EGFR phosphorylation is insufficient to say the EGFR is not phosphorylated. Figure 4b demonstrates that the kinase activity of the EGFR is required for STAT3 phosphorylation so it is difficult to assume that the kinase is active but the autophosphorylation ability of the receptor is impaired after only testing two tyrosines.

4) Also in Figure 4b, 20 μ m of gefitinib is used to treat MDAMB231 cells with STAT3 phosphorylation measured after 24 hr. 20 μ m of gefitinib is over twice the EC50 for MDAMB231 cells and all other STAT3 phosphorylation studies in this manuscript were performed within 30 min. It is unclear as to why the high dose of inhibitor was used for such a long period of time for this experiment.

5) When referring to Figure 5 it is stated that wt-EGFR and V948R-EGFR to not co-immunoprecipitate, however, it appears as though that combination was not tested in the experiments shown.

6) The interpretation of the western blots in Extended Data 8 does not match the line graph in Figure 6. Perhaps the individual tumors could be color coded to associate with the number of the sample indicated?

Overall, there are some interesting data here. There were some key words used to interpret the data that were not supported by the figures shown.

Response to Reviewers

Reviewer #1 (Remarks to the Author):

The authors have investigated the mechanistic basis for lack of efficacy of EGFR dimerization-blocking antibodies. They find monomeric EGFR-STAT3 signaling is oncogenic in TNBC and must be overcome therapeutically. Evidence of the tumor suppressive role of TMEM25 from the genetically engineered mouse models is convincing and striking. Biochemical evidence of the physical interaction between TMEM25 and EGFR is robust. The manuscript is exceptionally well written and clear. The mutation studies are flawed and need improvement or need to be removed. From what I can tell, there is no justification or need to study the mutation of TMEM as a mechanism of its inactivation in TNBC. The AAV experiments are exciting (here the mutation is appropriate to use as a 'negative control' experimentally).

Q1. In the introduction and perhaps even abstract, please state the % of TNBC patients with active EGFR signaling axis.

Response: As suggested by the reviewer, we stated the % of TNBC patients with overexpression of EGFR in the revised abstract (which had already been stated in the introduction in the previous version). Current studies have clearly demonstrated EGFR is important for TNBC progression and is associated with poor outcome of TNBC patients; however, to the best of our knowledge, there is no clear conclusion about the % of TNBC patients with active EGFR signaling axis yet. Failure of the clinical trial using EGFR antibody to target the EGFR dimerization might even lead to a conclusion that canonical EGFR signaling is not critical for TNBC progression. From our study, we know that the monomeric EGFR-STAT3 signaling axis has to be included to state the % of TNBC patients with active EGFR signaling axis, which surely needs to be examined in the future.

Q2. For lung metastasis findings in figure 2 and supplementary figure 4 – can authors demonstrate there is an effect on metastasis independent of an effect on primary tumor growth? Or perhaps metastasis is only affected because primary tumor growth rate is altered. One approach would be to normalize lung metastasis burden to primary tumor burden.

Response: Following the reviewer's suggestion, we normalized the numbers of lung metastasis nodules to the weights of primary tumor. Indeed, knockout or overexpression of TMEM25 significantly affected the rate of lung nodules to the weight of primary tumors (revised supplementary Fig. 4f). In fact, many studies have demonstrated that STAT3 signaling promotes tumor metastasis by activating Epithelial-to-mesenchymal transition (EMT) process, which is in good agreement with our finding that TMEM25 functions as a suppressor of EGFR/STAT3 signaling.

Q3. Figure 2- what is the EGFR status of MMTV pyMT mice in the tumor cells themselves? Is it expressed at the protein level? What is relative amount of monomeric vs dimeric EGFR? Is EGFR physically interacting with TMEM25 in the tumor cells growing in the mice?

Response: The expression of EGFR in the tumors from MMTV-PyMT mice had been

shown in Fig. 3d due to the flow of the text, which indicated that the EGFR is expressed in these tumors and its expression level is not affected by the amount of TMEM25. Technically, it is hard for us to treat tumor tissues with crosslinker to examine the dimer formation of EGFR, we therefore used collagenase IV to treat tumor tissues and collected the cells for the examination. However, because the EGFR protein level in the tumor cells from MMTV-PyMT mice is much lower than that in MDA-MB-231 cells (Fig. for reviewers 1a), we could not effectively detect the dimer formation in tumor cells from MMTV-PyMT mice using crosslinker. Nevertheless, we were able to demonstrate that nearly all EGFR in MDA-MB-231 cells exist as monomers in the serum starvation condition used in our analysis and treatment of EGF substantially induced EGFR dimer formation (revised Supplementary Fig. 8e, f). In addition, we proved that EGFR interacts with TMEM25 in the tumor cells from MMTV-PyMT mice by Co-IP assay (revised Supplementary Fig. 4i).

Q4. In human TNBC, is the typical ratio of monomeric vs dimeric EGFR known? Does this correlate with any disease parameters or outcomes?

Response: There are only a few studies reported a ratio of monomeric vs dimeric EGFR in cells among over a hundred thousand papers in PubMed searched by using EGFR, and most of these studies used overexpressed fluorescent protein-tagged but not endogenous EGFR. To the best of our knowledge, there is no literature reported a ratio of monomeric vs dimeric EGFR in TNBC yet.

Q5. In the introduction, please more clearly communicate the knowledge gap this study addresses. EGFR is known to activate STATs. Is it understood whether that is monomeric or dimerized EGFR? Clearly state the Y2H screen identified a novel (?) interaction between EGFR and TMEM25.

Response: As suggested by the reviewer, we added “It has been widely believed that dimerization is an essential step for activation of EGFR” in the revised introduction. The previous studies for EGFR-mediated STATs activation did not clarify whether it was through EGFR dimer or monomer, although they were obviously assumed through EGFR dimer based on current understanding of EGFR activation mechanism. Our study, for the first time, demonstrated that monomeric EGFR can activate downstream signaling. Therefore, it is hard for us to clearly state whether the previous studies for EGFR-mediated STATs activation is through monomeric or dimeric EGFR in the introduction. Meanwhile, we added “novel” to state the interaction between EGFR and TMEM25 in the beginning of result section because this interaction has not been reported yet.

Q6. Is the mechanism of physical interaction between EGFR and TEMEM25 possible to be disrupted with an antibody or steric inhibitor? It would be good if authors have data to address this. At a minimum this possibility should be discussed.

Response: Theoretically, it is possible to disrupt the interaction between EGFR and TMEM25 using an antibody or steric inhibitor. However, disrupting the interaction between EGFR and TMEM25 will result in an abnormal activation of STAT3 signaling to promote tumor progression. Therefore, it will not be beneficial to block the interaction between EGFR and TMEM25. Rather than disrupting the interaction

between EGFR and TMEM25, blocking the interaction between EGFR and STAT3 might be a potential strategy for targeted therapy of TNBC, which we had already discussed in the end of the discussion section.

Q7. In the TNBC cell lines used, does TMEM25 level correlate with proliferation, tumorigenicity, etc?

Response: We compared the proliferation, survival in serum starvation, and colony formation of the TNBC cell lines used in our experiments (Fig. for reviewers 1b-d). As shown in Fig. for reviewers 1d, these TNBC cell lines have dramatic differences in various signaling pathways. For example, 4T1 cells have less amount of EGFR, but their AKT and ERK are constitutively activated; MDA-MB-231 cells have relative less amount of TMEM25, and their ERK is constitutively activated. Hence, it is hard for us to establish a simple correlation between TMEM25 levels and cell growth rates in these cell lines due to the complexity in regulation of cell growth by various signaling pathways.

Q8. Please edit this sentence for clarity: “we found that TMEM25 protein levels were markedly decreased in 23 out of 28 samples and showed...” – decreased relative to what? Do you mean expression was low? How was low vs high judged? In this same sentence, it is stated there is negative correlation with p-STAT3, suggesting ‘TMEM is required for restraining STAT3 activation to suppress TNBC progression.’ The correlation does not in fact suggest this sweeping finding. Please remove that phrase.

Response: We appreciate the reviewer’s strictness to our expression. We added “compared with that in the matched adjacent normal tissues” and removed “suggesting that TMEM25 is required for restraining STAT3 activation to suppress human TNBC progression” to make our conclusion more accurate.

Q9. The TMEM25 mutation studies are inherently flawed.

a. In text please report what % (and the raw number of patients) with the G59D TMEM25 mutation. Is this mutation expected to be somatic? Nonsynonymous? SIFT/polyphen etc prediction of deleterious impact on protein function?

b. Why was the G59D mutation not examined experimentally, since apparently this is the only one detected in a substantial number of TNBC samples?

c. The non-TNBC mutations may be completely irrelevant to TNBC. The expression of these mutations in TNBC is not justified, so the experiment in which they are introduced into 231 or 4t1 cells does not make much sense. Authors need to discretely state this in the results section or remove those experiments altogether.

Response: We agree that the mutation studies of TMEM25 do not fit the theme of this manuscript very well. We only found the G59D mutation exists in TNBC by checking cancer (COSMIC) database. There is no percentage report for this mutation. We did examine its effect on inhibiting STAT3 activation together with other mutations (previous Extended Data Fig. 9a); however, it did not show significant difference on inhibiting STAT3 activation compared with wild-type TMEM25, which is likely due to locating in the extracellular domain. We hence removed most of the mutation studies and only kept R326W and L338F as negative controls in AAV experiments.

Q10. The AAV experiment in which WT TMEM inhibited pyMT tumor growth is exciting. The mutation construct here can be evaluated as a tool to inactivate TMEM but again it must be stated very clearly there is no evidence for this mutation being relevant in the TNBC space.

Response: Following the reviewer's suggest, we used the two TMEM25 mutants (R326W and L338F) as negative controls and clearly stated that theses 2 mutants were not found in TNBC in the revised manuscript.

Q11. Have authors tested STAT3 inhibition in mice +/-TMEM? If not, please discuss. This would be a good experiment.

Response: As suggested by the reviewer, we examined the effects of knockdown of STAT3 on tumor growth in *TMEM25^{+/+}* and *TMEM25^{-/-}* mice. As predicted, knockdown of STAT3 drastically inhibited the tumor growth in both *TMEM25^{+/+}* and *TMEM25^{-/-}* mice (revised Supplementary Fig. 7), indicating that STAT3 signaling is essential for MDA-MB-231 cells growing in mice.

Minor comments

Q1. In text there are a couple instances where it is stated that IHC was done on some tumors and readers are referred to an extended data figure. Authors should also include a statement in the text summarizing the result of the IHC experiment.

Response: As suggested by the reviewer, we added statements in the revised text to summarize the results of the IHC experiments.

Q2. In the introduction please discuss the status of STAT inhibition in TNBC preclinically and clinically.

Response: As suggested by the reviewer, we briefly summarized the status of preclinical and clinical studies of STAT3 inhibition in cancer treatment in the revised introduction.

Q3. Please cite the PiggyBac approach.

Response: The reference for PiggyBac approach has been cited in the revised manuscript.

Q4. Methods- please provide catalog numbers and concentrations for EGF and all inhibitors. Please provide supplemental data or citation justifying the choice of dosages used.

Response: The catalog numbers and conditions for treatment of EGF or different inhibitors were provided in the revised methods according to the reviewer's suggestion. The references for the treatment conditions were also cited.

Q5. Methods- IP section is too simple and lacks citation. Authors should provide sufficient detail so that readers can accurately repeat the experiment.

Response: As suggested by the reviewer, more details and reference were provided for IP assay in the revised methods.

Q6. Extended fig 5- figure and legend should indicate what the IHC antibody is

recognizing. Is it Ki67 again?

Response: The IHC assay in Extended Data Fig. 5c was for phosphorylated STAT3. Sorry that we missed the label during organizing the figures. The label is now added in the revised Supplementary Fig. 5c, and is indicated in the figure legend.

Q7. Why have authors not done IHC for TMEM25 especially in the patient samples? Is there a suitable antibody for IHC?

Response: We have tried all commercially available and our lab-created polyclonal TMEM25 antibodies for IHC. Unfortunately, we could not find any one suitable for IHC assay. We are planning to generate monoclonal antibodies using our TMEM25 knockout mice and wish we can get a good one for IHC. However, we have not got what we want yet. Hence, we are unable to examine TMEM25 by IHC currently.

Q8. Extended fig 6- some x axis labels are missing.

Response: We apologize that we missed the x axis labels in the Extended Data Fig. 6e-g. The labels were added in the revised figures.

Reviewer #2 (Remarks to the Author):

This is a generally well-written and fairly comprehensive manuscript outlining the molecular and functional role of TMEM25 in tumor formation, and in negatively regulating the ability of monomeric EGFR to activate STAT3. This is accomplished by physical complexing with EGFR.

I have only a few minor comments:

Q1. The terms "upregulated" and "downregulated" are used a little imprecisely. Please state specifically "downregulated" relative to what? There are data relative to normal mammary epithelium and there are data relative to other tumor types. Please specify which comparison is made where in the paper to clarify for the reader.

Response: According to the reviewer's suggestion, we clarified the relative standards for words "upregulated" and "downregulated" used where could be confusing in the revised manuscript.

Q2. Given the vagueness of the expression statements, it would be of interest to see the expression pattern in the normal human (or even just mouse) mammary epithelium relative to tumor by immunostaining rather than bulk RNA.

Response: We totally agree with the reviewer that examining the expression of TMEM25 in tissues by immunostaining is a better approach. Unfortunately, although we have tried all available TMEM25 antibodies (commercial and lab-created), we were unable to get an antibody suitable for immunostaining.

Q3. Some additional in vivo work, perhaps in PDX, or at least orthotopic transplantation of the manipulated cell lines (subcutaneous is well known to show different results in many cases than the inguinal (#4) mammary fat pad). It is possible that the results would be somewhat different, or perhaps even significantly different.

Response: The MDA-MB-231 cells were subcutaneously injected in to nude mice

because relative large amount of cells are needed to form tumors, which is hard to orthotopically injected into mammary fat pad. However, for the transplant tumor using 4T1 cell line, the cells were actually orthotopically injected into mammary fat pad of BALB/c mice (Figure legend for Supplementary Fig. 3d-f). Moreover, we have collaborated with Dr. Shiyong Guo from GemPharmatech Co., Ltd. (Nanjing, China) to examine the effect of providing TMEM25 by AAV on inhibiting tumor growth in a PDX TNBC tumor model (revised Fig. 6f-h; revised Supplementary Fig. 10e, f).

Reviewer #3 (Remarks to the Author):

The epidermal growth factor receptor (EGFR) has been shown to be overexpressed in the majority of triple negative breast cancers (TNBCs), however, its role in the initiation and progression of these cancers remains elusive. Specific to this manuscript, monoclonal antibodies targeting the EGFR family of kinases have not shown promise in the clinic. The authors of this manuscript provide evidence that the EGFR does not respond to monoclonal antibody inhibition in TNBC because the EGFR is signaling as a ligand-independent monomer and thereby does not require ligand binding or dimerization, the mechanism of monoclonal antibody inhibition. They have further evidence that this is mediated by downregulation of the transmembrane protein TMEM25 which at normal expression levels binds to monomeric EGFR and prevents ligand-independent signaling. The authors continue to characterize the ligand-independent monomeric signaling of EGFR to occur through STAT3 but not through MAPK or AKT.

While these results have added a new player, TMEM25, into the EGFR signaling field and have the potential to generate great potential in the treatment of TNBC, there are several key experiments missing from this manuscript.

Q1. The presence of EGFR monomers and the absence of autocrine ligand production are not biochemically confirmed in this manuscript. The presence of monomers and absence of dimers of the EGFR by a technique such as cross-linking followed by non-reducing gel electrophoresis would help to strengthen this major conclusion. It has been published that MDAMB231 cells produce several EGFR ligands in an autocrine fashion. It would strengthen the ligand-independent conclusion of this paper to demonstrate that they are not being produced or binding to EGFR in your model system.

Response: Following the reviewer's suggestion, we used crosslinker disuccinimidyl suberate (DSS) to examine dimer formation of EGFR under different conditions. As predicted, there was certain amount of EGFR dimer in MDA-MB-231 cells growing in 10% FBS; however, there was very little EGFR dimer in cells under serum starvation. As positive control, treatment of EGF drastically induced dimer formation of EGFR (revised Supplementary Fig. 8e, f). Hence, it is plausible that most of EGFR were monomers in our experiments under serum starvation. Meanwhile, we examined the amount of EGF under different conditions as well. The levels of EGF in 10% FBS and serum starvation condition were very low compared with that in EGF treatment condition (Fig. for reviewers 1e). Hence, it is very likely that most EGFR in serum starved MDA-MB-231 were in ligand free monomer status.

Q2. When TMEM25 is knocked out and overexpressed in the key tumor growth

experiments in Figure 2, there are no companion figures to demonstrate the changes in expression, either in the cells prior to injection in the mouse or in the subsequent mouse tumors. There is one figure, Extended Figure 3 that shows one example of knock out in the cell line and one in the tumor but it is not clear with which experiment those data associate. I understand that these are CRISPR knockouts but the level of TMEM25 expression endogenously in these cells is almost undetectable. For the knockout to have such a profound biological change in cell and tumor growth is perplexing, particularly when the overexpression had much a higher expression but does not have the same significance with respect to biological impact. This is particularly key with the transgenic models (Extended Figure 4f). There are no detectable differences via Western blot between the knockout and wild-type TMEM25 expression.

Response: The Supplementary Fig. 3c (named Extended Fig. 3c in previous version) presented the expression levels of TMEM25 in cells grown *in vitro*, and the Supplementary Fig. 3g showed the TMEM25 expression in the tumors. We added more details to the figure legend for Supplementary Fig. 3c to clarify this. The TMEM25 antibody detects a non-specific band at the position of TMEM25 for tissue samples, likely because it is a polyclonal antibody. Therefore, we re-performed the experiment for the previous Extended Fig. 4f. We treated the tumor tissues using collagenase and collected the cells for IB assay, which provided us a much clearer result (revised Supplementary Fig. 4g).

Q3. Only looking at two sites of EGFR phosphorylation is insufficient to say the EGFR is not phosphorylated. Figure 4b demonstrates that the kinase activity of the EGFR is required for STAT3 phosphorylation so it is difficult to assume that the kinase is active but the autophosphorylation ability of the receptor is impaired after only testing two tyrosines.

Response: To address this point, we exogenously expressed wild-type EGFR and EGFR-V948R mutant in EGFR knockout cells, immunoprecipitated EGFR and examined the tyrosine phosphorylation using a phospho-Tyr specific antibody pY20. As shown in revised Supplementary Fig. 8d, exogenously expressed wild-type EGFR had a basal tyrosine phosphorylation, and EGF treatment substantially increased tyrosine phosphorylation of EGFR. On the contrary, EGFR-V948R did not show any tyrosine phosphorylation no matter with or without EGF treatment. Hence, this result clearly demonstrated that V948R mutation totally abolished the auto-phosphorylation of tyrosine in EGFR.

Q4. Also in Figure 4b, 20 μM of gefitinib is used to treat MDAMB231 cells with STAT3 phosphorylation measured after 24 hr. 20 μM of gefitinib is over twice the EC50 for MDAMB231 cells and all other STAT3 phosphorylation studies in this manuscript were performed within 30 min. It is unclear as to why the high dose of inhibitor was used for such a long period of time for this experiment.

Response: We chose 20 μM and 24 hours treatment of gefitinib for MDA-MB-231 cells based on 2 literatures (Sordella R, et al. *Science* 2004, 305(5687):1163-7; El Guerrab A, et al. *Oncotarget* 2016, 7(45):73618-73637). In the first one, the authors treated the NSCLC cells 72 hours with different doses of gefitinib, and the cells with

wild-type EGFR were strongly affected only when the concentration reached 20 μM . The second literature reported a condition for treatment of MDA-MB-231 cells with different doses of gefitinib for 24 hours. Similarly, treatment with a concentration of 20 μM or above gave a robust effect on inhibiting cell growth. They also reported that the IC50 of gefitinib for MDA-MB-231 was 16.5 μM . Meanwhile, we did test the conditions of gefitinib treatment for MDA-MB-231 cells in our preliminary experiment and found 24 hours treatment with 20 μM gefitinib can give a good inhibition on EGFR activity. The 30 min we used in our experiments were all for treatment of EGF. EGF treatment will induce a fast response of EGFR, usually in minutes, so that we tested time points of 0, 5, 15, and 30 minutes. The inhibitors were used to inhibit the basal activity of EGFR toward STAT3 in the absence of TMEM25 in our experiment, which might not be as fast as the response of EGFR to EGF treatment. We know that small molecule inhibitors may cause nonspecific effects. Therefore, we also strengthened the conclusion by using sh-RNAs to knockdown these kinases (revised Fig. 4a, b, previous Extended Data Fig. 6a,b), which gave similar results obtained by using inhibitors.

[REDACTED]

Adapted from Sordella R, et al. *Science* 2004, 305:1163

[REDACTED]

Adapted from El Guerrab A, et al. *Oncotarget* 2016, 7:73618

Q5. When referring to Figure 5 it is stated that wt-EGFR and V948R-EGFR to not co-immunoprecipitate, however, it appears as though that combination was not tested in the experiments shown.

Response: The sentence we used in the text “As previously reported, we confirmed that EGFR-V948R was not able to form dimer as wild-type EGFR did by Co-IP assay (Fig. 5a)” was trying to say that wild-type EGFR could form dimers, but EGFR-V948R could not. We were not trying to test the interaction between wild-type EGFR and EGFR-V948R mutant. To avoid potential misunderstanding, we revised this sentence as “Consistent with previous report², we confirmed that exogenously overexpressed wild-type EGFR could form dimer, whereas EGFR-V948R could not (Fig. 5a)”.

Q6. The interpretation of the western blots in Extended Data 8 does not match the line graph in Figure 6. Perhaps the individual tumors could be color coded to associate with the number of the sample indicated?

Overall, there are some interesting data here. There were some key words used to interpret the data that were not supported by the figures shown.

Response: Following the reviewer’s suggestion, we used different colors and symbols to indicate the No. of tumor samples (revised Fig. 6a).

REVIEWER COMMENTS

Reviewer #2 (Remarks to the Author):

The authors failed to respond adequately to my critique. Simply not wanting to do any more experimentation does not justify not addressing reviewer critiques.

Q1: Some instances of vague use of "upregulated" and "downregulated" are fixed, but not all. This is very important. There are instances in the literature in which an "upregulated" gene in tumors was actually "downregulated" relative to the normal tissue of origin. One can say "across tumors, X is more highly expressed in this subset relative to that" but the up and down regulation terms cannot be used like that.

Q2: TMEM25 Immunostaining in the normal breast to demonstrate "downregulation" in cancer, or even just a normal mouse mammary gland (this was the out if the authors for whatever reason could not obtain normal breast) was requested. The authors respond to this by saying that they tried several antibodies but they did not work. Yet in the text of the paper they state:

"Interestingly, in breast cancers, TMEM25 mRNA levels were most dramatically downregulated in TNBCs compared with normal mammary tissues (Supplementary Fig. 2c)". Thus, there was clearly no issue obtaining tissue for the experiment.

They also state:

"To verify this interaction in cells, we performed coimmunoprecipitation (Co-IP) assays using either exogenously expressed or endogenous EGFR and TMEM25, confirming that EGFR could indeed interact with TMEM25 (Fig. 1a, b and Supplementary Fig. 1a, b). Meanwhile, we determined that EGFR and TMEM25 were mainly colocalized at plasma membrane by immunofluorescence assay (Fig. 1c).

If one can do immunostaining with a TMEM antibody, why didn't the authors respond to the question by conducting the necessary experiment, which they demonstrate that they can do. Either that or the data presented with respect to TMEM25 is artifactual if the antibodies truly do not work.

Another way to demonstrate "downregulation" at the protein level would be to conduct the IP in normal tissue, which again they state that the antibody needed for this works.

Finally, one could conduct IP-MS in a pinch.

Q3: I requested that Some additional in vivo work, perhaps in PDX, or at least orthotopic transplantation of the manipulated cell lines (subcutaneous is well known to show different results in many cases than the inguinal (#4) mammary fat pad). It is possible that the results would be somewhat different, or perhaps even significantly different, or in fact, opposite. A literature search will demonstrate this point.

In response, the authors state: "The MDA-MB-231 cells were subcutaneously injected in to nude mice because relative large amount of cells are needed to form tumors, which is hard to orthotopically injected into mammary fat pad."

This is grossly incorrect on both points: 1) MDA-MB-231 cells are among the most aggressive breast cancer cells around. In the literature, transplantation by limiting-dilution transplantation of 231 cells grown in vivo has shown that one can use as few as 10 cell to grow tumors in the mouse mammary fat pad. Similar data are in the literature for cells grown in vitro. A literature search would be all that is required. 2) the mouse mammary gland is NOT difficult to inject into. It is one of the most accessible organs in the animal, and cells can be injected into the #4 mammary gland with a Hamilton syringe directly through the skin or via a small (2mm) incision over the location of the #4 gland.

This strikes me as a very poor excuse especially when the very next sentence in the response directly contradicts the above explanation by stating: " However, for the transplant tumor using 4T1 cell line, the cells were actually orthotopically injected into mammary fat pad of BALB/c mice (Figure legend for Supplementary Fig. 3d-f).

There is no technical difference between injection of human cells vs. mouse cells. The authors obviously have the ability to conduct the experiment. They should do so to address my concern.

These are not difficult or overly time-consuming experiments to do, and they are not unreasonable to expect in response to a reviewer

Reviewer #3 (Remarks to the Author):

Thank you for working diligently to respond to the questions by the reviewers. I have one remaining concern; the mouse immunoblotting work found in supplemental figure 9 and quantified in figure 6a. Thank you for adding the colors and legend to make the comparison feasible. I understand from experience that these experiments are notoriously difficult, however, I just cannot match up the results seen in the immunoblots with the lines in the figures. These are key results for translation of the data. Perhaps providing a table with the numbers you generated for the quantifications? I feel like some lines should be going down that are going up and vice versa. When you determined your negative correlation between TMEM25 and pSTAT3 did you pool the data or look at each tumor sample pair as a set?

Response to Reviewers

Reviewer #2 (Remarks to the Author):

The authors failed to respond adequately to my critique. Simply not wanting to do any more experimentation does not justify not addressing reviewer critiques.

Q1: Some instances of vague use of "upregulated" and "downregulated" are fixed, but not all. This is very important. There are instances in the literature in which an "upregulated" gene in tumors was actually "downregulated" relative to the normal tissue of origin. One can say "across tumors, X is more highly expressed in this subset relative to that" but the up and down regulation terms cannot be used like that.

Response: Following the reviewer's request, we fixed all "upregulated" and "downregulated" in our manuscript.

Q2. TMEM25 Immunostaining in the normal breast to demonstrate "downregulation" in cancer, or even just a normal mouse mammary gland (this was the out if the authors for whatever reason could not obtain normal breast) was requested. The authors respond to this by saying that they tried several antibodies but they did not work. Yet in the text of the paper they state:

"Interestingly, in breast cancers, TMEM25 mRNA levels were most dramatically downregulated in TNBCs compared with normal mammary tissues (Supplementary Fig. 2c)". Thus, there was clearly no issue obtaining tissue for the experiment.

They also state:

"To verify this interaction in cells, we performed coimmunoprecipitation (Co-IP) assays using either exogenously expressed or endogenous EGFR and TMEM25, confirming that EGFR could indeed interact with TMEM25 (Fig.1a, b and Supplementary Fig. 1a, b). Meanwhile, we determined that EGFR and TMEM25 were mainly colocalized at plasma membrane by immunofluorescence assay (Fig. 1c).

If one can do immunostaining with a TMEM antibody, why didn't the authors respond to the question by conducting the necessary experiment, which they demonstrate that they can do. Either that or the data presented with respect to TMEM25 is artifactual if the antibodies truly do not work.

Another way to demonstrate "downregulation" at the protein level would be to conduct the IP in normal tissue, which again they state that the antibody needed for this works.

Finally, one could conduct IP-MS in a pinch.

Response: The results in Supplementary Fig. 2c was obtained by mining the public datasets from the cancer genome atlas (TCGA) using the UALCAN platform (<http://ualcan.path.uab.edu/analysis.html>), as we mentioned in the main text ("We

analyzed the datasets from the cancer genome atlas (TCGA) using the UALCAN platform (<http://ualcan.path.uab.edu/analysis.html>) and found that....”) and in the figure legend. The only clinical samples we collected were the 28 TNBC samples with their surrounding normal tissues, which we used to examine the TMEM25 protein levels to confirm that the protein levels of TMEM25 in TNBC tumor tissues are indeed lower than that in the surrounding normal tissues (Supplementary Fig. 9).

Because the current available antibodies for TMEM25 are all polyclonal antibodies, they gave very high background when doing immunostaining or immunohistochemistry assays. As shown in the Figure for reviewers 1, although the difference of TMEM25 expression levels between wild-type and TMEM25 knockout cells or tumor samples could be determined by Western blotting, the background raised by TMEM25 polyclonal antibody totally blocked the signal of TMEM25 in immunostaining or immunohistochemistry assays. We are sorry that we did not clearly state the immunofluorescence experiment done in Fig. 1c was using GFP-tagged TMEM25 in the main text (although it was described in the figure legend), and it was GFP but not TMEM25 antibody that was used for immunostaining. We have re-addressed it in the revised manuscript.

We were luckily able to examine the protein levels of TMEM25 by Western blotting assay using TMEM25 polyclonal antibody by separating TMEM25 from the nonspecific proteins that could be recognized by the polyclonal TMEM25 antibody through SDS-PAGE, which cannot be achieved for immunostaining or immunohistochemistry assays. Therefore, all the protein levels of TMEM25 in our samples, including the mouse tumor samples and clinical TNBC samples, were examined by Western blotting assay in our manuscript. And, since we were able to examine protein levels of TMEM25 by Western blotting assay, we don't think it is necessary to conduct IP-MS to confirm the protein levels of TMEM25, which unlikely can give more accurate results than that done by Western blotting assay.

Q3. I requested that Some additional in vivo work, perhaps in PDX, or at least orthotopic transplantation of the manipulated cell lines (subcutaneous is well known to show different results in many cases than the inguinal (#4) mammary fat pad). It is possible that the results would be somewhat different, or perhaps even significantly different, or in fact, opposite. A literature search will demonstrate this point.

In response, the authors state: "The MDA-MB-231 cells were subcutaneously injected in to nude mice because relative large amount of cells are needed to form tumors, which is hard to orthotopically injected into mammary fat pad."

This is grossly incorrect on both points: 1) MDA-MB-231 cells are among the most aggressive breast cancer cells around. In the literature, transplantation by limiting-dilution transplantation of 231 cells grown in vivo has shown that one can use as few as 10 cell to grow tumors in the mouse mammary fat pad. Similar data are in the literature for cells grown in vitro. A literature search would be all that is required. 2) the mouse mammary gland is NOT difficult to inject into. It is one of the most

accessible organs in the animal, and cells can be injected into the #4 mammary gland with a Hamilton syringe directly through the skin or via a small (2mm) incision over the location of the #4 gland.

This strikes me as a very poor excuse especially when the very next sentence in the response directly contradicts the above explanation by stating: " However, for the transplant tumor using 4T1 cell line, the cells were actually orthotopically injected into mammary fat pad of BALB/c mice (Figure legend for Supplementary Fig. 3d-f).

There is no technical difference between injection of human cells vs. mouse cells. The authors obviously have the ability to conduct the experiment. They should do so to address my concern.

These are not difficult or overly time-consuming experiments to do, and they are not unreasonable to expect in response to a reviewer.

Response: Sorry that we misunderstood the reviewer's request for "some additional in vivo work, perhaps in PDX, or". We did examine the effect of TMEM25 provision in inhibiting TNBC tumor growth in PDX mouse model as the reviewer suggested (Fig. 6f-h, and Supplementary Fig. 10e,f), but we did not redo the MDA-MB-231 tumor formation experiment. We now have reperfomed the MDA-MB-231 tumor formation experiments by orthotopically injecting the cells into mammary fat pad as the reviewer requested (revised Fig. 2c-e, Supplementary Fig. 3g,h, Supplementary Fig. 5c, and Supplementary Fig. 7b-e).

Reviewer #3 (Remarks to the Author):

Thank you for working diligently to respond to the questions by the reviewers. I have one remaining concern; the mouse immunoblotting work found in supplemental figure 9 and quantified in figure 6a. Thank you for adding the colors and legend to make the comparison feasible. I understand from experience that these experiments are notoriously difficult, however, I just cannot match up the results seen in the immunoblots with the lines in the figures. These are key results for translation of the data. Perhaps providing a table with the numbers you generated for the quantifications? I feel like some lines should be going down that are going up and vice versa. When you determined your negative correlation between TMEM25 and pSTAT3 did you pool the data or look at each tumor sample pair as a set?

Response: We really appreciate the carefulness of the reviewer. We apologize that we had mislabeled the samples #1-#4 and #9-#12 due to the accidental switch of gel-1 and gel-3 when we did quantification (Figure for reviewers 2). The protein levels in Western blotting assay were quantified and the data of all the pairs of tumors and surrounding normal tissues were inputted into GraphPad Prism 8 software and computed for nonparametric Spearman correlation by the software program. Following the reviewer's suggestion, we provided the actual numbers for the quantification in an excel file (Source Data Figure 6a). In addition, we replaced the previous Figure 6a (in which

relative pSTAT3 levels were calculated using pSTAT3/ β -tubulin) using a new one with relative pSTAT3 levels that were calculated using pSTAT3/total STAT3, which we now think is more appropriate.

Figure for reviewers 1. Application of TMEM25 polyclonal antibody in immunofluorescence, immunohistochemistry, and immunoblotting assays.

a, b, Immunofluorescent staining (**a**) and immunoblotting (**b**) of TMEM25 in *TMEM25*^{+/+} and *TMEM25*^{-/-} MDA-MB-231 cells. Scale bar indicates 100 μ m.

c-e, Immunofluorescent staining (**c**), immunohistochemistry (**d**), and immunoblotting (**e**) of TMEM25 in representative tumors from *TMEM25*^{+/+} and *TMEM25*^{-/-} *MMTV-PyMT* mice. Scale bar indicates 100 μ m.

f-h, Immunofluorescent staining (**f**), immunohistochemistry (**g**), and immunoblotting (**h**) of TMEM25 in a representative human clinical TNBC sample and its surrounding normal tissue. Scale bar indicates 100 μ m.

Figure for reviewers 2. Correction of the mislabeling.

REVIEWERS' COMMENTS

Reviewer #4 (Replacement reviewer for Reviewers #2 and #3, Remarks to the Author):

As a newly assigned reviewer, I have first carefully read the paper and then evaluated the rebuttal letter response to reviewers. Overall, the experimental data is solid and supported by comprehensive analyses. The comments from both Reviewers 2 and 3 have been adequately addressed.